# Online Ad Procurement in Non-stationary Autobidding Worlds

**Jason Cheuk Nam Liang**
MIT
jcnliang@mit.edu

**Haihao Lu**
University of Chicago
haihao.lu@chicagobooth.edu

**Baoyu Zhou**
University of Michigan
zbaoyu@umich.edu

## Abstract

Today's online advertisers procure digital ad impressions through interacting with autobidding platforms: advertisers convey high level procurement goals via setting levers such as budget, target return-on-investment, max cost per click, etc. Then ads platforms subsequently procure impressions on advertisers' behalf, and report final procurement conversions (e.g. click) to advertisers. In practice, advertisers may receive minimal information on platforms' procurement details, and procurement outcomes are subject to non-stationary factors like seasonal patterns, occasional system corruptions, and market trends which make it difficult for advertisers to optimize lever decisions effectively. Motivated by this, we present an online learning framework that helps advertisers dynamically optimize ad platform lever decisions while subject to general long-term constraints in a realistic bandit feedback environment with non-stationary procurement outcomes. In particular, we introduce a primal-dual algorithm for online decision making with multi-dimensional decision variables, bandit feedback and long-term uncertain constraints. We show that our algorithm achieves low regret in many worlds when procurement outcomes are generated through procedures that are stochastic, adversarial, adversarially corrupted, periodic, and ergodic, respectively, without having to know which procedure is the ground truth. Finally, we emphasize that our proposed algorithm and theoretical results extend beyond the applications of online advertising.

## 1 Introduction

Automated bidding (or autobidding for short) has become the dominant mode for advertisers to procure digital ad inventories and impressions, contributing to more than \$120 billion dollar ad spend in 2022 and more than $90\%$ of total online ad transaction volumes [2, 1]. In an autobidding platform, advertisers only need to convey their high-level procurement goals for an ad campaign to the platform, which then takes charge of procuring ads on advertisers' behalf. Such procurement goals are communicated to a platform through platform *levers*, which are adjustable parameters that advertisers can control to influence the bidding process and campaign performance. To exemplify, Figure 1 displays certain several levers presented on the Google Ads interface, where an advertiser can set per-campaign budgets, target cost-per-actions, campaign duration, campaign schedules, targeting, etc; similar examples are also shown in related literature [21].

As the primary avenue for advertisers to run ad campaigns on autobidding platforms and influence ad conversion outcomes (e.g., clicks), making efficient lever decisions is vital to advertisers to achieve their procurement objectives. However, advertisers face many challenges in practice when optimizing for lever decisions, namely high-dimensional decision making under long-term constraints, non-stationary autobidding environments, and limited procurement feedback.

**High-dimensional decision making and long-term constraint satisfaction.** Making multiple lever decisions involves evaluating numerous possible combinations of lever configurations, which is computationally intensive and time-consuming in real-time decision making setups. Also, advertisers need to understand the potential interactions and dependencies between levers, as adjusting one lever may have unintended consequences or interactions with other levers, making it challenging to

37th Conference on Neural Information Processing Systems (NeurIPS 2023).

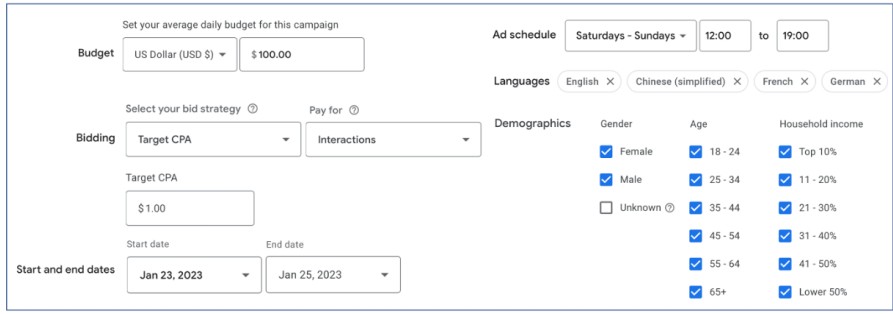

Figure 1: Example levers on the Google Ads interface to create digital advertising campaigns.

predict the overall impact of adjustments accurately. Furthermore, advertisers may need to satisfy certain long-term constraints over time, e.g., limiting total spend under a budget, or achieving certain return-on-investment targets. Hence in addition to analyzing the interactions between different levers, advertisers also need to concern long-term consequences of making certain lever decisions.

**Limited feedback on procurement outcome and constraints.** Despite the fact that autobidding via lever decisions greatly simplifies advertisers' ad procurement process as they no longer have to handle the intricacies of procuring individual ad impressions, the procurement procedure becomes a black box for advertisers, as advertisers only have limited visibility into the specific details of how the platform executes ad placement processes. This lack of transparency can make it challenging for advertisers to control the nuances of their campaign execution through lever decisions, and it amplifies the complexity to conduct counterfactual analyses on outcomes for past lever decisions.

**Many non-stationary autobidding worlds.** Autobidding procurement environments are highly non-stationary, as a wide spectrum of latent factors in online ad marketplaces may greatly vary procurement outcomes of the same lever decisions in different time periods. These latent factors include but are not limited to changing user preferences, seasonality effects, shifts in market trends, occasional malfunctions in autobidding platforms, etc. These dynamics may influence how users interact with different types of ads, which necessitates continuous adjustment of lever decision strategies to adapt to current and future market conditions.

To address these challenges, in this work we answer the following questions: *How should an advertiser dynamically set multiple levers to optimize conversion outcomes subject to long-term constraints under limited information? And can she run a unified algorithm that can perform well while being agnostic to many types of non-stationary autobidding procurement environments?*

Motivated by these questions, in this work we study an advertiser's online high-dimensional lever decision problem with long-term constraints under limited bandit feedback in many non-stationary worlds. We summarize our main contributions as followed:

**1. Modelling online lever decisions in many worlds using online constrained optimization with bandit function-valued feedback and uncertain constraints (Section 2).** We cast the advertiser lever decision problem as an online constrained optimization with bandit function-valued feedback and uncertain constraints, where functions at which (lever) decisions are evaluated correspond to the conversion and constraint functions in our autobidding setup. Further, we model real-world non-stationarity in autobidding environments as possibly time-varying distributions from which conversion and constraint functions are sampled, and then further applied to lever decisions. Under this model, we discuss five different input procedures from which the sequence of functional distributions are sampled to model stochastic, adversarial, corrupted, periodic, and ergodic environments; see Section 2.2 for more details. To the best of our knowledge, from a modelling perspective this is the first work to model a high-dimensional lever decision problem in practical non-stationary autobidding worlds.

**2. Proposing a constrained bandit optimization algorithm universally applicable to many worlds (Section 3).** We develop a unified bandit optimization algorithm with robust performance guarantees across different worlds. Our algorithm incorporates four key designs to handle bandit function-valued feedback and unknown non-stationary environments. 1. we utilize dual descent to update dual variables associated with long-term constraints and decouple decisions over time; 2. we employ a primal-ascent-based bandit online convex optimization (BOCO) approach to make (primal) lever decisions to cope with bandit function valued feedback; 3. we implement an exponential weights

expert algorithm on top of primal-ascent BOCO, where each expert corresponds to a different primal ascent step size. This enables our algorithm to adapt to the optimal primal ascent step size for each world without prior knowledge on which world we are in; 4. our algorithm dynamically checks for realized constraint violations and applies "safe lever decisions" to ensure long-term constraint satisfaction. For further details, we refer the readers to Algorithm 1 in Section 3.

**3. Analyzing the performance of proposed algorithms in many worlds (Section 4).** We present theoretical analysis (Theorem 4.2) on the regret bound of the proposed algorithm, and we show that our unified algorithm can achieve reasonable regret bounds with all five input procedures. These regret bounds are summarized in the following Table 1.

| Stochastic | $\delta$-corrupted | Adversarial | Periodic | Ergodic |
|---|---|---|---|---|
| $\mathcal{O}(T^{\frac{3}{4}})$ | $\mathcal{O}(T^{\frac{3}{4}} + \delta)$ | $\left(1 - \frac{1}{\xi}\right) \mathrm{OPT}(\mathcal{P}_{1:T})$ $+ \tilde{\mathcal{O}}\left(A \cdot T^{\frac{3}{4}}\right)$ | $\mathcal{O}(T^{\frac{3}{4}} + q\sqrt{T})$ | $\mathcal{O}(T^{\frac{3}{4}} + \kappa\sqrt{T})$ |

Table 1: $T$-period regret bounds for Algorithm 1 under different input models. Here $A > 0$ is some algorithm dependent factor which we will later specify in Section 4.

The parameters $(\delta, \xi, \mathrm{OPT}(\mathcal{P}_{1:T}), q, \kappa)$ are formally defined in Section 2.2 and Theorem 4.2. Finally, in Section 5 we remark all results are applicable to other problems.

## 1.1  Related works.

**Autobidding.** In an autobidding framework, there is a considerable body of works that study the price of anarchy, which aims to improve worst case individual or total advertiser welfare guarantees w.r.t. the optimal welfare via mechanism design frameworks; see e.g. [23, 7, 20, 38, 22]. We remark that this line of work does not concern developing online learning algorithms. On the other hand, numerous works have concentrated on developing online bidding algorithms for repeated ad auctions [41, 10, 31, 30], as well as designing repeated selling mechanisms for ad impression allocation; see [12, 27] and references therein. However, as discussed in the introduction, autobidding platforms conduct bidding on behalf of advertisers while keeping bidding and selling mechanism details undisclosed. In this study, we treat the bidding procedure and selling mechanism as a black box and directly model bidding and auction outcomes using conversion functions; see Section 2. To the best of our knowledge, the most pertinent work to this paper is [21], which explores a similar ad procurement problem by optimizing levers within a bandit environment. However, this work solely focuses on the stochastic world and optimizing a single lever (i.e., a 1-dimensional decision space). In contrast, our paper develops a unified algorithm capable of making high-dimensional lever decisions in many worlds.

**Online convex optimization.** In Section 2, we cast the advertiser's online learning problem of interest to a high-dimensional bandit online convex optimization problem with uncertain long-term constraints, and develop an algorithm that yields good performance guarantees under different procedures from which the objective and constraint functions are generated. There has been a rich line of works that study bandit convex optimization with no long-term constraints in stochastic and adversarial worlds [25, 32, 42, 17, 44], as well as works that study (full-information feedback) online convex optimization with long-term constraints [36, 33, 43, 35]. Further, works that study both bandit feedback and long-term constraints either consider single-dimensional (such as multi-arm bandits with constraints) [40, 6, 4, 24], or consider static regret (i.e., benchmarking performance to that of a single optimal action) [13, 16, 14, 37]. This paper distinguishes itself from these two streams of works by considering high-dimensional decisions as well as dynamic regret, i.e., comparing to the best sequence of actions instead of a single action; see Eq. (1). Finally, all aforementioned works only study algorithms in stochastic or adversarial setups, whereas in this work we go beyond these two worlds and address more complex learning environment such as periodic, corrupted, ergodic, and finite switching. To the best of our knowledge, the only related work that develops a universal algorithm under "many world" setups is [10]. However, [10] considers a full-information scenario where the online decisions in period can be made after observing realized objective and constraint functions during that period. In this work, we present an efficient algorithm to handle bandit feedback.

**Notation.** For any vector $z \in \mathbb{R}^d$, let $\|z\|$ be its Euclidean norm. Denote $\mathbb{B} = \{z \in \mathbb{R}^d : \|z\| \leq 1\}$ as the $d$-dimensional unit ball centered at $\mathbf{0}$, and let $\mathbb{S} = \{z \in \mathbb{R}^d : \|z\| = 1\}$ be the unit sphere. For any set $\mathcal{S}$, let $U(\mathcal{S})$ be the uniform distribution over $\mathcal{S}$. Let $e$ denote the vector whose components

are all 1's, and $e_j$ be the unit vector whose $j$'th position is 1. We use $\mathcal{O}$ notation to represent the asymptotic order of a term when the period $T \to \infty$ and ignore the $\log T$ terms.

## 2 Autobidding with bandit feedback in many worlds (input models)

### 2.1 Autobidding as a bandit online optimization with long-term uncertain constraints

Consider an advertiser repeatedly interacting with an ad platform over $T \in \mathbb{N}$ periods, where each period can be interpreted as a single ad campaign that is run on the platform. During each period $t \in [T]$, the advertiser makes $d \geq 1$ lever decisions denoted as $x_t \in \mathcal{X} \subseteq \mathbb{R}^d$, e.g., setting the per-campaign budget, campaign duration, per-campaign target return-on-investment, and max spend per conversion; see Figure 1 for example lever decisions in practice. Here, $\mathcal{X}$ is some compact and convex decision set whose diameter is $D = \sup_{\{x, x'\} \subseteq \mathcal{X}} \|x - x'\|$. For simplicity, assume $\mathbf{0} \in \mathcal{X}$ so $\|x\| \leq D$ for any $x \in \mathcal{X}$. After the lever decisions $x_t$ are made the campaign is fully executed via autobidding, the advertiser observes bandit feedback for her campaign outcomes: she only observes her realized conversion $f_t(x_t) \in \mathbb{R}_{\geq 0}$ (e.g., number of clicks on her ads), as well as some constraint balance $g_t(x_t) \in \mathbb{R}^K$. The conversion and constraint functions $(f_t, g_t)$ are sampled from some (possibly infinite) support $\mathcal{S}$ according to distribution $\mathcal{P}_t \in \Delta(\mathcal{S})$ (we will discuss how $\mathcal{P}_t$'s are generated by nature in Section 2.2). Using the notation $\mathcal{P}_{1:T} = (\mathcal{P}_t)_{t \in [T]}$ the advertiser's hindsight optimization problem is

$$\text{OPT}(\mathcal{P}_{1:T}) = \max_{x \in \mathcal{X}^T} \sum_{t \in [T]} \mathbb{E}_{\mathcal{P}_t}[f_t(x_t)] \text{ s.t. } \sum_{t \in [T]} \mathbb{E}_{\mathcal{P}_t}[g_{k,t}(x_t)] \geq 0 \quad k = 1 \ldots K . \tag{1}$$

Here, we use a constraint function $g_{k,t} : \mathcal{X} \to \mathbb{R}$ to characterize general performance metrics of the advertiser for her ad campaigns. In the following, we present several examples for constraint functions that are widely used in practice or studied in literature. For illustrative purposes, we assume $x_{t,1} = e_1^\top x_t$ (i.e., the first lever) represents the per-campaign budget for the $t$'th campaign.

- *Long-term budget constraint.* The advertiser has a total budget $BT > 0$. Then by letting constraint function $g_{k,t}(x) = B - x_{t,1}$, we have $\sum_{t \in [T]} g_{k,t}(x_t) \geq 0 \implies \sum_{t \in [T]} x_{t,1} \leq BT$, which means total campaign spent over $T$ periods (assuming each campaign fully depletes the campaign budget) must be less than the advertiser's total budget $BT$.

- *Long-term return-on-investment constraint.* The advertiser intends to safeguard a long-term return-on-investment $\gamma > 0$, i.e., she attains a long-term average of at least $\gamma$ conversions per dollar spent. Then by considering $g_{k,t}(x) = f_t(x) - \gamma x_{t,1}$, we have $\sum_{t \in [T]} g_{k,t}(x_t) \geq 0 \implies \sum_{t \in [T]} f_t(x_t) \geq \gamma \sum_{t \in [T]} x_{t,1}$, which means total conversion over $T$ periods is at least $\gamma$ times total spend.

Finally, we make the following mild assumptions on conversion and constraint functions.[1]

**Assumption 2.1** (Mild assumptions on conversion and constraint functions). *For any $(f, g) \in \mathcal{S}$, $f$, $g_1 \ldots g_K$ are all bounded concave functions, i.e., we assume $\sup_{x \in \mathcal{X}} \|g(x)\|_\infty \leq \bar{G}$ and $\sup_{x \in \mathcal{X}} |f(x)| \leq \bar{F}$ for some $\bar{G}, \bar{F} < \infty$. Moreover, $f$ and $g$ are $L$-Lipschitz, i.e., for any $\{x, x'\} \subseteq \mathcal{X}$, we have $|f(x) - f(x')| \leq L\|x - x'\|$ and $\|g(x) - g(x')\| \leq L\|x - x'\|$. Furthermore, there exists $(f, g) \in \mathcal{S}$ such that $\min_{x \in \mathcal{X}} \min_{k \in [K]} g_k(x) < 0$.*

### 2.2 Five input models characterizing many autobidding worlds

In this subsection, we describe structural properties of the input distribution sequence $\mathcal{P}_{1:T}$, and shed light on how we utilize various properties to model a wide spectrum of autobidding environments (called worlds) such as time-varying user preferences, seasonality, shifts in market trends, etc., that may potentially lead to different procurement outcomes for the same ad campaign lever decisions.

**Stochastic**: There exists some probability distribution $\mathcal{P} \in \Delta(\mathcal{S})$ such that $\mathcal{P}_1 = \ldots = \mathcal{P}_T = \mathcal{P}$. This stochastic world represents a stationary autobidding environment where the underlying latent factors influencing user behaviors (and correspondingly conversion results) remain constant over time; see [31, 9, 21].

---

[1]These assumptions are justified in many related works in autobidding; see [21, 14] and references therein. For example, [21] shows that the conversion function is concave and piecewise-linear when autobidding platforms procure ad impressions on behalf of advertisers in standard second price or VCG auctions.

**δ-corrupted.** There exists $\mathcal{P} \in \Delta(\mathcal{S})$ as well as $\delta \in \mathbb{N}$ periods $\mathcal{T} = \{\tau_1 \ldots \tau_\delta\} \subset [T]$ such that $\mathcal{P}_t = \mathcal{P}$ for all $t \notin \mathcal{T}$. This δ-corrupted input sequence represents occasional anomalies in the autobidding environment that may be caused by systematic malfunctions in the autobidding platform, or deliberate attempts by malicious competitors to exploit the system for their own benefit; e.g., some competitors may engage in click fraud to inflate the number of clicks on our ads to exhaust our budget or generate false data to manipulate autobidding algorithms (for example, see [29]).

**Adversarial.** $\mathcal{P}_{1:T}$ is adversarially chosen by nature before the process starts, and the distributions over time can possibly be non-identical and/or dependent. This adversarial world can be viewed a hypothetical extreme case for the δ-corrupted world where each period the procurement outcomes can potentially be corrupted by an adversary. Adversarial input sequences have been widely studied in the literature to assess algorithmic performances in worst-case scenarios; see [3, 30].

**Periodic.** There exists period length $q \in \mathbb{N}$ such that $T = cq$ for some integer $c \geq 2$ with $\mathcal{P}_{1:T}$ satisfying $\mathcal{P}_{1:q} = \mathcal{P}_{q+1:2q} = \cdots = \mathcal{P}_{(c-1)q+1:T}$. This periodic world captures regular cyclic patterns or fluctuations in the user behavior over specific time intervals; e.g., seasonality, day-of-week patterns, and time of a day.

**Ergodic.** $\mathcal{P}_{1:T}$ is an ergodic process (e.g., an irreducible and aperiodic Markov chain or stationary autoregressive processes), where there exists some $\kappa > 0$ and a stationary distribution such that the distance between $\kappa$-step transition probabilities and this stationary distribution decreases exponentially fast in $\kappa$. Mathematically, given the input distribution sequence $\mathcal{P}_{1:T}$, denote $\mathcal{P}_{(t+\kappa)|[t-1]}$ as the conditional distribution of $(f_{t+\kappa}, \boldsymbol{g}_{t+\kappa})$ conditioned on the realizations $\{(f_\tau, \boldsymbol{g}_\tau)\}_{\tau \in [t]}$. Then, for the ergodic input model there exists a stationary distribution $\widetilde{\mathcal{P}} \in \Delta(\mathcal{S})$ and absolute constant $R > 0$ such that

$$\sup_{\{(f_t, \boldsymbol{g}_t)\}_{t \in [T]} \in \mathcal{S}^T} \sup_{t \in [T-\kappa]} \|\mathcal{P}_{(t+\kappa)|[t-1]} - \widetilde{\mathcal{P}}\|_{TV} \leq \delta := R \exp(-\kappa) ; \tag{2}$$

We remark that exponentially decaying ergodic models have been studied widely in the context of online optimization; see e.g. [10, Section 5.2] for further discussions. To make the problem of interest tractable, we assume $\kappa > \log(T)$. Intuitively, an ergodic input sequence signifies that the procurement outcomes in close time proximity are correlated, which is commonly observed in real-world autobidding systems, as they often involve iterative processes that enable procurement algorithms (operated on behalf of advertisers) to converge to a stable state; see details in [9, 28].

### 2.3 Minimizing regret subject to long-term constraints with a universal algorithm

In this work, we take the perspective of an advertiser making repeated lever decisions as described in Section 2.1. We focus on designing a single online algorithm that determines a lever decision $\boldsymbol{x}_t \in \mathcal{X}$ in each period $t$ with the goal to minimize regret $\mathcal{R}_T$ (defined as follows) under any input sequence $\mathcal{P}_{1:T}$ while satisfying long-term constraints

$$\mathcal{R}_T = \mathrm{OPT}(\mathcal{P}_{1:T}) - \sum_{t \in [T]} \mathbb{E}\left[f_t(\boldsymbol{x}_t)\right] \text{ and } \sum_{t \in [T]} \mathbb{E}\left[\boldsymbol{g}_t(\boldsymbol{x}_t)\right] \geq \boldsymbol{0}. \tag{3}$$

Here, the expectation is taken w.r.t. randomness from the input sequence as well as any randomness in our algorithm. We highlight that our desired policy should be agnostic to the input model as well as the input sequence $\mathcal{P}_{1:T}$. We remark that our work distinguishes itself from related work that study solely stochastic and/or adversarial input models, as we aim to design a universal algorithm that achieves salient performance over any input model described in Section 2.2.

## 3 A universal constrained BOCO framework for many worlds

In this section, we focus on designing an algorithm that achieves low regret (per Eq. (3)) while maintaining long-term constraint satisfaction in any world described in Section 2.2, without having to know which world we are in. We first highlight three main challenges for our problem of interest.

**Dynamic benchmark lever sequence and unknown input sequence $\mathcal{P}_{1:T}$.** Recall the hindsight optimal problem in Eq. (1), with which we are comparing our algorithm's performance to the optimal sequence of lever decisions over time given any input sequence $\mathcal{P}_{1:T}$, instead of comparing to a single optimal lever decision as in many related works; e.g., see [15] and references therein. This dynamic optimal sequence presents a very strong benchmark and makes learning very difficult as we (ideally) need to account for variations in the underlying non-stationary ground truth input sequence $\mathcal{P}_{1:T}$. Nevertheless, we do not know the input sequence $\mathcal{P}_{1:T}$ in advance, nor do we know in which world the sequence belongs (Section 2.2).

**Bandit function-value (zeroth-order) feedback.** Our setup concerns a bandit function-value feedback model, where the realized conversion and constraint functions in each period, namley $f_t : \mathcal{X} \to \mathbb{R}_{\geq 0}$ and $g_t : \mathcal{X} \to \mathbb{R}^K$ are never revealed to us, and we can only access their function values at our single lever decisions during the period. We also contrast our setup with a "two-point feedback" setup in online convex optimization literature, where in each period one can access function values twice (see [44] and references therein). Using the context of our setup, in a "two-point feedback" after nature samples $f_t : \mathcal{X} \to \mathbb{R}_{\geq 0}$ and $g_t : \mathcal{X} \to \mathbb{R}^K$ from $\mathcal{P}_t$, we can observe the values for two lever decisions $\{\boldsymbol{x}_t^{(1)}, \boldsymbol{x}_t^{(2)}\} \subseteq \mathcal{X}$, namely $(f_t(\boldsymbol{x}_t^{(1)}), \boldsymbol{g}_t(\boldsymbol{x}_t^{(1)}))$ and $(f_t(\boldsymbol{x}_t^{(2)}), \boldsymbol{g}_t(\boldsymbol{x}_t^{(2)}))$.

**Universal algorithm for many worlds.** Instead of developing separate algorithms that cope with each individual world described in Section 2.1, we aim to develop a single universal algorithm that performs well in every world without exploiting any knowledge on the realized input sequence $\mathcal{P}_{1:T}$ or its structural properties.

The key component of our algorithm to handle the aforementioned challenges is the following Lagrangian function: for any $f_t : \mathcal{X} \to \mathbb{R}_{\geq 0}$ and $\boldsymbol{g}_t : \mathcal{X} \to \mathbb{R}^K$, let's define $\mathcal{L}_t : \mathcal{X} \times \mathbb{R}_{\geq 0}^K \to \mathbb{R}$ such that

$$\mathcal{L}_t(\boldsymbol{x}, \boldsymbol{\lambda}) = f_t(\boldsymbol{x}) + \boldsymbol{\lambda}^\top \boldsymbol{g}_t(\boldsymbol{x}) . \tag{4}$$

Surrounding this Lagrangian function, our proposed algorithm is formally stated in Algorithm 1 with four key components described as follows.

**1. Dual descent to decouple decisions across time.** As our lever decisions over time should be somewhat intertwined with one another due to the presence of long term constraints, we decouple this cross-period dependency by lagrangifying the hindsight problem in Eq. (1) w.r.t. some benchmark dual variables. In each period $t$, we maintain an estimate of dual variables $\boldsymbol{\lambda}_t \in \mathbb{R}_{\geq 0}^K$ corresponding to each of our $K$ constraints, and adopt a standard dual descent approach to dynamically adjust these estimates; see Eq. (9). In particular, after observing $(f_t(\boldsymbol{x}_t), \boldsymbol{g}_t(\boldsymbol{x}_t))$, we update our dual variable by projecting each coordinate of $\boldsymbol{\lambda}_t - \eta \nabla_{\boldsymbol{\lambda}} \mathcal{L}_t(\boldsymbol{x}_t, \boldsymbol{\lambda}_t) = \boldsymbol{\lambda}_t - \eta \boldsymbol{g}_t(\boldsymbol{x}_t)$ onto some interval $[0, \frac{\bar{F}}{\beta}]$ where $\eta > 0$ is the dual descent step size, $\bar{F}$ is the maximum achievable conversion, and $\beta > 0$ is some "safety buffer" to be defined later.

**2. Primal ascent to handle bandit function-valued feedback in many worlds.** Given a dual variable $\boldsymbol{\lambda}_t$ during period $t$, standard optimization frameworks suggest optimizing primal decisions (i.e., lever decisions) by setting $\boldsymbol{x}_t = \arg\max_{\boldsymbol{x} \in \mathcal{X}} \mathbb{E}_{\mathcal{P}_t}[\mathcal{L}_t(\boldsymbol{x}, \boldsymbol{\lambda}_t)]$. However, in our setting this is not possible due to the bandit function-valued feedback structure: we do not know $f_t(\cdot)$, $\boldsymbol{g}_t(\cdot)$ nor $\mathcal{P}_t$, and hence cannot optimize for $\mathbb{E}_{\mathcal{P}_t}[\mathcal{L}_t(\cdot, \boldsymbol{\lambda}_t)]$. To handle this, we view the primal decision problem as a *bandit online convex optimization problem (BOCO)* where the adversarial objective functions are $\mathcal{L}_t(\cdot, \boldsymbol{\lambda}_t)$, and run a BOCO algorithm to make primal lever decisions in each period.

Mathematically, we define the BOCO objective function $h_t : \mathcal{X} \to \mathbb{R}$ as

$$h_t(\boldsymbol{x}) = \mathcal{L}_t(\boldsymbol{x}, \boldsymbol{\lambda}_t) = f_t(\boldsymbol{x}) + \boldsymbol{\lambda}_t^\top \boldsymbol{g}_t(\boldsymbol{x}) . \tag{5}$$

Then, the BOCO algorithm constructs an estimate $\boldsymbol{\nabla}_t$ of $\boldsymbol{\nabla} h_t(\boldsymbol{x})$ based on a random perturbation approach with perturbation parameter $\rho > 0$ (see Eq. (7) and [25] for more details in random perturbations methods to estimate gradients). Next, the BOCO algorithm computes primal decision $\boldsymbol{x}_t$ by an ascent type update, i.e., projecting $\boldsymbol{x}_{t-1} + \gamma \boldsymbol{\nabla}_t$ back onto $\mathcal{X}$, where $\gamma > 0$ is the primal ascent step size; see Eq. (9). The loss we incur by running BOCO primal update instead of setting $\arg\max_{\boldsymbol{x} \in \mathcal{X}} \mathbb{E}_{\mathcal{P}_t}[h_t(\boldsymbol{x}_t)]$ in the full information setting depends on the specific type of world we are in, i.e., the input sequence $\mathcal{P}_{1:T}$. We later show in Lemma 4.3 that input sequences in each world induce some comparator sequence $\boldsymbol{y}_1 \ldots \boldsymbol{y}_T \in \mathcal{X}$, so the conversion loss due to BOCO primal ascent updates can be characterized as some BOCO "dynamic" regret $\sum_{t \in [T]} h_t(\boldsymbol{y}_t) - h_t(\boldsymbol{x}_t^{BOCO})$. Later in Lemma 4.6, we show that our BOCO algorithm achieves low dynamic regret against any comparator sequence $\boldsymbol{y}_1 \ldots \boldsymbol{y}_T \in \mathcal{X}$.

**3. Choosing primal ascent step sizes from expert advice for different worlds.** Different worlds (and correspondingly different BOCO comparator sequences) may require different "optimal" BOCO primal ascent step sizes in order to achieve optimal regret guarantees. Since we are unaware of the world from which the input sequence $\mathcal{P}_{1:T}$ is generated, on top of our primal ascent procedure we run a meta "expert" algorithm that allows us to adaptively approximate optimal step sizes in each world. In particular, we consider $N$ experts, each corresponding to a primal ascent step size $\gamma_i > 0$,

and produce independent lever decision based on primal ascent with her own step size; see Eq. (9). Then, we dynamically maintain exponential weights $\boldsymbol{w}_t \in \Delta_N$ in the $N$-dimensional simplex over $N$ experts, based on their past performance measured by some surrogate loss function defined in Eq. (8). Finally we set the weighted average primal decision over all experts as our ultimate primal decision; see Eq. (6). We later show in Lemma A.3 that such exponentially weighted average BOCO primal ascent decisions achieve low conversion loss compared to the "optimal" BOCO primal ascent expert (i.e., step size) in every world, without having to know which world we are in.

**4. Constraint violation check to ensure constraint satisfaction.** To ensure long-term constraint satisfaction, we maintain a constraint balance $B_{k,t} = \sum_{\tau \in [t-1]} g_{k,t}(\boldsymbol{x}_\tau)$ for each constraint $k \in [K]$ that keeps track of our deviation from satisfying the constraint. To make our problem tractable, we make the assumption that there is always a "safe action" with which we are always guaranteed to attain some small positive constraint balance.[2]

**Assumption 3.1** (Safe action). *Assume there exists some $\bar{\beta} > 0$ and an action $\widetilde{\boldsymbol{x}}_\beta \in \mathcal{X}$ such that for any $(f, \boldsymbol{g}) \in \mathcal{S}$, we have $g_k(\widetilde{\boldsymbol{x}}_\beta) > \bar{\beta}$ for all $k \in [K]$.*

With the constraint balance, in each period before making a lever decision, we check the following: if by playing the safe action in all future rounds nearly violates constraints (step 2 in Algorithm 1), then we hard stop our algorithm and play the safe action in all subsequent rounds. We remark that we do not necessarily know the constraint function value lower bound $\bar{\beta} > 0$ in Assumption 3.1. However, we show later in Theorem 4.1 that by considering any safety buffer $\beta < \bar{\beta}$ (e.g., taking $\beta = 1/\log(T)$ for large enough $T$), we are hard stopping more conservatively, and thus maintain constraint satisfaction. Note that our final regret scales with $\frac{1}{\beta^2}$; see Lemma 4.6.

---

**Algorithm 1**

---

**Input:** Initial dual variable $\boldsymbol{\lambda}_1 = \boldsymbol{0}$, primal expert solutions $\widetilde{\boldsymbol{x}}_1^i = \boldsymbol{0}$ for any $i \in [N]$; perturbation parameter $\rho > 0$ and $\alpha \in (0,1)$; primal ascent step sizes for experts $\{\gamma_1, \ldots, \gamma_N\} > 0$; dual descent step size $\eta$; learning rate of the meta-algorithm $\epsilon$; initial expert weights $w_{i,1} = 1$ for all $i \in [N]$; safety buffer $\beta > 0$.
1: Initialize constraint balance $B_{k,1} = 0$ for all $k \in [K]$
2: **while** $\{$for all $k \in [K]$, $B_{k,t} - \bar{G} + \beta(T - t - 1) \geq 0\}$ **do**
3:    **Compute exponentially weighted average forecaster**:

$$\widetilde{\boldsymbol{x}}_t = \frac{1}{\sum_{i \in [N]} w_t^i} \sum_{i \in [N]} w_t^i \widetilde{\boldsymbol{x}}_t^i \qquad (6)$$

4:    Sample $\boldsymbol{u}_t \sim U(\mathbb{S})$ uniformly at random from the unit sphere.
5:    Set $\boldsymbol{x}_t = \widetilde{\boldsymbol{x}}_t + \rho \boldsymbol{u}_t$ and observe $f_t(\boldsymbol{x}_t)$ and $\boldsymbol{g}_t(\boldsymbol{x}_t)$. Update $B_{k,t+1} = B_{k,t} + g_{k,t}(\boldsymbol{x}_t)$.
6:    **Construct gradient estimate for** $\nabla_{\boldsymbol{x}} \mathcal{L}(\boldsymbol{x}_t, \boldsymbol{\lambda}_t)$

$$\boldsymbol{\nabla}_t = \frac{d}{\rho} \left( f_t(\boldsymbol{x}_t) + \boldsymbol{\lambda}_t^\top \boldsymbol{g}_t(\boldsymbol{x}_t) \right) \cdot \boldsymbol{u}_t \qquad (7)$$

7:    **Update exponential weights for experts**: Let $\ell_t : \mathcal{X} \to \mathbb{R}$ be a surrogate loss function to measure the performance of each expert. Then we update expert weights by

$$w_{i,t+1} = w_{i,t} \exp\left( -\epsilon \ell_t(\widetilde{\boldsymbol{x}}_t^i) \right) \quad \text{where} \quad \ell_t(\boldsymbol{z}) = \boldsymbol{\nabla}_t^\top (\widetilde{\boldsymbol{x}}_t - \boldsymbol{z}) \qquad (8)$$

8:    **Primal ascent per expert and dual descent**:

$$\widetilde{\boldsymbol{x}}_{t+1}^i = \Pi_{(1-\alpha)\mathcal{X}}(\widetilde{\boldsymbol{x}}_t^i + \gamma_i \boldsymbol{\nabla}_t) \text{ and } \boldsymbol{\lambda}_{t+1} = \Pi_{[0, \frac{\bar{F}}{\beta} \boldsymbol{e}]}(\boldsymbol{\lambda}_t - \eta \nabla_{\boldsymbol{\lambda}} \mathcal{L}_t(\boldsymbol{x}_t, \boldsymbol{\lambda}_t))_+ \qquad (9)$$

9: **end while**
10: Set stopping time $\tau_A = t$. For $t = \tau_A \ldots T$ set safety option $\boldsymbol{x}_t = \widetilde{\boldsymbol{x}}_\beta$.

---

## 4 Performance analysis of our constrained BOCO algorithm

In this section, we analyze the performance of Algorithm 1 under input sequence $\mathcal{P}_{1:T}$ which may be generated from any world described in Section 2.2. Our first result Lemma 4.1 shows that our algorithm maintains long-term constraint satisfaction almost surely for any input sequence.

---

[2]The existence of a safe action is not unnatural, and is studied in many literature; see [24, 21, 14]. For instance, let's recall our examples for constraint functions in Section 2.1 and assume we only have a long-term budget constraint. Then the corresponding safe action for the constraint function $g_{k,t}(\boldsymbol{x}_t) = B - x_{t,1}$ would be any $\boldsymbol{x}_t$ whose first entry $x_{t,1} = 0$ so that we acquire positive constraint value $B > 0$.

**Lemma 4.1** (Strict constraint satisfaction). *Suppose Assumption 3.1 hold, and $T$ is large enough so that the safety buffer $\beta = \frac{1}{\log(T)} < \bar{\beta}$, where $\bar{\beta}$ is defined in Assumption 3.1. Then, for any $k \in [K]$, we have $\sum_{t \in [T]} g_{k,t}(\boldsymbol{x}_t) > 0$.*

The main result of this paper is the following Theorem 4.2, where we bound the regret of our proposed Algorithm 1 in all worlds specified in Section 2.2.

**Theorem 4.2** (Bounding regret in many worlds). *Let the safety buffer $\beta = \frac{1}{\log(T)}$, and the number of experts $N = \max\left(1, \left\lceil -\log_2\left(K^{-\frac{1}{6}}(1+DT)^{\frac{1}{2}}T^{-\frac{3}{4}}\right)\right\rceil + 1\right)$, where $K$ is the total number of constraints and $D$ is diameter of the decision set $\mathcal{X}$. Choose the corresponding primal ascent step sizes for the $N$ experts as $\{\gamma_1 \ldots \gamma_N\} = \{2^{-i}K^{-\frac{1}{6}}(1+DT)^{\frac{1}{2}}T^{-\frac{3}{4}} : i = 0 \ldots N-1\}$. Then by taking dual descent step size $\eta = \frac{1}{\sqrt{KT}}$, random perturbation parameter $\rho = K^{\frac{1}{3}}T^{-\frac{1}{4}}$ and $\alpha = \min\left(\frac{1}{2}, K^{\frac{1}{3}}T^{-\frac{1}{4}}\right)$, and exponential weighted expert learning rate $\epsilon = T^{-\frac{1}{2}}$, we obtain the following bounds on $\mathcal{R}_T$ in each input setting of interest: (1) **Stochastic:** $\mathcal{R}_T \leq \mathcal{O}(T^{3/4})$; (2) **Adversarial:** $\mathcal{R}_T \leq \left(1 - \frac{1}{\xi}\right)\mathrm{OPT}(\mathcal{P}_{1:T}) + \tilde{\mathcal{O}}\left(\sqrt{1 + P(\widetilde{\boldsymbol{y}}_{1:T})} \cdot T^{\frac{3}{4}}\right)$, where $\xi = 1 - \frac{1}{\bar{\beta}}\min_{(f,\boldsymbol{g})\in\mathcal{S}}\min_{k\in[K],\boldsymbol{x}\in\mathcal{X}} g_k(\boldsymbol{x}) > 1$ under Assumption 2.1, $\bar{\beta} > 0$ is defined in Assumption 3.1, $\widetilde{\boldsymbol{y}}_t = \arg\max_{\boldsymbol{x}\in\mathcal{X}} f_t(\boldsymbol{x}_t) + \boldsymbol{\lambda}_t^\top \boldsymbol{g}_t(\boldsymbol{x}_t)$, and $P(\widetilde{\boldsymbol{y}}_{1:T}) = \sum_{t\in[T-1]}\|\widetilde{\boldsymbol{y}}_t - \widetilde{\boldsymbol{y}}_{t+1}\|$; (3) $\delta$-**corrupted:** $\mathcal{R}_T \leq \mathcal{O}(T^{3/4} + \delta)$; (4) **Periodic:** $\mathcal{R}_T \leq \mathcal{O}(T^{3/4} + q\sqrt{T})$; (5) **Ergodic:** $\mathcal{R}_T \leq \mathcal{O}(T^{3/4} + \kappa\sqrt{T})$.*

We also summarize our regret bounds in Table 1. Although it is always challenging to identify which input procedure the data currently comes from, Theorem 4.2 shows that the proposed Algorithm 1 achieves reasonable regret bound in five input settings without knowing which setting we are in. We remark that in the adversarial setting, our algorithm is $(1 - \frac{1}{\xi})$-competitive, namely, it can achieve at least a $\frac{1}{\xi}$-portion of the reward compared to OPT; later in Section 5 we comment that no sublinear regret is achievable. For the $\delta$-corrupted world, we corrupt $\delta$ instances in the input sequence over time, so if $\delta = 0$, we recover the stochastic world and achieve diminishing $\mathcal{O}(T^{3/4})$ regret; in the case we corrupt all samples, then we recover the adversarial case, and the regret becomes $\mathcal{O}(T)$ as expected. In the periodic world, if period length $q = 1$, we recover the stochastic case.

**Remark 4.1.** *We remark that our regret upper bound in the adversarial case depends on the dual variables $\{\boldsymbol{\lambda}_t\}_{t\in[T]}$ generated throughout the algorithm, which contrasts adversarial regret bounds in [44] that only depend on the total variance of the optimal sequence of 1. This is due to that fact that [44] handles a non-constrained online optimization problem, whereas we tackle a more difficult problem with long-term adversarial constraints that leads to more severe regret degradation as we ensure satisfaction of long-term adversarial constraints. We acknowledge that this bound should ideally be independent of dual variables, and we leave this for future work.*

In the rest of this section, we present a proof scratch of Theorem 4.2, and the formal proof of the results can be found in Appendix A. The high-level idea of the proof is that we decompose total regret into three components as in Lemma 4.3 bellow, namely the conversion losses due to hard stopping, dual descent, and primal ascent, respectively. Then, we bound each component's regret.

**Lemma 4.3** (Regret decomposition). *Let $\left(\boldsymbol{\lambda}_t \in \mathbb{R}^K_{\geq 0}\right)_{t\in[T]}$ be the sequence of dual variables generated from Algorithm 1, and define $h_t(\boldsymbol{x}) = \mathcal{L}_t(\boldsymbol{x}, \boldsymbol{\lambda}_t)$ where there Lagrangian function $\mathcal{L}_t(\boldsymbol{x}, \boldsymbol{\lambda})$ is defined in Eq. (4). Then we have*

$$\mathrm{OPT}(\mathcal{P}_{1:T}) - \sum_{t\in[T]}\mathbb{E}\left[f_t(\boldsymbol{x}_t)\right] \leq \bar{F}(T - \tau_A) + \sum_{t\in[\tau_A]}\boldsymbol{\lambda}_t^\top \boldsymbol{g}_t(\boldsymbol{x}_t) + \mathcal{R}_{\mathrm{BOCO}}(\tau_A) \qquad (10)$$

*where $\mathcal{R}_{\mathrm{BOCO}}(\tau_A)$ admits the following bounds in each input setting: (1) **Stochastic:** $\max_{\boldsymbol{x}\in\mathcal{X}}\sum_{t\in[\tau_A]}h_t(\boldsymbol{x}) - h_t(\boldsymbol{x}_t)$; (2) **Adversarial:** $\sum_{t\in[\tau_A]}h_t(\widetilde{\boldsymbol{y}}_t) - h_t(\boldsymbol{x}_t) + \left(1 - \frac{1}{\xi}\right)\mathrm{OPT}(\mathcal{P}_{1:T})$ where $\widetilde{\boldsymbol{y}}_t = \arg\max_{\boldsymbol{x}\in\mathcal{X}}\mathbb{E}_{\mathcal{P}_t}[f_t(\boldsymbol{x}) + \boldsymbol{\lambda}_t^\top \boldsymbol{g}_t(\boldsymbol{x})]$; (3) $\delta$-**corrupted:** $\max_{\boldsymbol{x}\in\mathcal{X}}\sum_{t\in[\tau_A]}h_t(\boldsymbol{x}) - h_t(\boldsymbol{x}_t) + \mathcal{O}(\delta)$; (4) **Periodic:** $\max_{\boldsymbol{x}\in\mathcal{X}}\sum_{t\in[\tau_A]}h_t(\boldsymbol{x}) - h_t(\boldsymbol{x}_t) + \mathcal{O}(\eta q T)$; (5) **Ergodic:** $\max_{\boldsymbol{x}\in\mathcal{X}}\sum_{t\in[\tau_A]}h_t(\boldsymbol{x}) - h_t(\boldsymbol{x}_t) + \mathcal{O}(\eta\kappa T)$.*

Here, we remark that the first term $\bar{F}(T - \tau_A)$ is the loss due to hard stopping to maintain long-term constraint satisfaction; the second term $\sum_{t\in[\tau_A]}\boldsymbol{\lambda}_t^\top \boldsymbol{g}_t(\boldsymbol{x}_t)$ is the loss due to dual descent; and the

final term $\mathcal{R}_{\text{BOCO}}(\tau_A)$ is the loss due to primal ascent. The first two components turn out to be identical in different settings, and the input model only affects the last term $\mathcal{R}_{\text{BOCO}}(\tau_A)$. We next present our key results with which we use to bound first two components (dual descent and hard stop loss), and the third component (primal ascent loss), respectively.

**Bounding dual descent and hard stop loss.** Our strategy to bound the loss due to dual descent and hard stopping relies on bounding some cumulative "complementary slackness" induced from dual descent in the following Lemma 4.4. We note that the bound in Lemma 4.4 holds for any $\boldsymbol{\lambda} \in [\mathbf{0}, \frac{\bar{F}}{\beta}\boldsymbol{e}]$, and thus by choosing appropriate $\boldsymbol{\lambda}$, we can "internalize" the losses due to dual descent and hard stopping, as demonstrted in Lemma 4.5.

**Lemma 4.4** (Bounding complementary slackness). *For any $\boldsymbol{\lambda} \in [\mathbf{0}, \frac{\bar{F}}{\beta}\boldsymbol{e}]$ and $t \in [T]$, we have* $\sum_{\tau \in [t]}(\boldsymbol{\lambda}_\tau - \boldsymbol{\lambda})^\top \boldsymbol{g}_\tau(\boldsymbol{x}_\tau) = \sum_{\tau \in [t]}(\boldsymbol{\lambda}_\tau - \boldsymbol{\lambda})^\top \nabla_{\boldsymbol{\lambda}}\mathcal{L}_\tau(\boldsymbol{x}_\tau, \boldsymbol{\lambda}_\tau) \leq \frac{\eta}{2}tK\bar{G}^2 + \frac{1}{2\eta}\|\boldsymbol{\lambda}\|_2^2$

**Lemma 4.5** (Internalizing dual descent and hard stop losses.). *Given the stopping time $\tau_A \in [T]$, w.p. 1 we have* $\bar{F}(T - \tau_A) + \sum_{t \in [\tau_A]}\boldsymbol{\lambda}_t^\top \boldsymbol{g}_t(\boldsymbol{x}_t) \leq \frac{\eta}{2}TK\bar{G}^2 + \frac{1}{2\eta}\left(\frac{\bar{F}}{\beta}\right)^2 + \bar{F} + \frac{\bar{F}}{\beta}\bar{G}$.

**Bounding primal ascent loss (dynamic BOCO regret).** We present our main result Lemma 4.6, which bounds the loss due to primal ascent for given any comparator sequence $\boldsymbol{y}_1 \ldots \boldsymbol{y}_T \in \mathcal{X}$.

**Lemma 4.6** (Bounding primal ascent (dynamic BOCO regret)). *For any $i \in [N]$, $t \in [T]$ and any sequence $\boldsymbol{y}_{1:t} \in \mathcal{X}^t$ we have* $\sum_{\tau \in [t]} h_\tau(\boldsymbol{y}_\tau) - h_\tau(\boldsymbol{x}_\tau) \leq \mathcal{O}\left(\frac{(\rho+\alpha)T}{\beta} + \frac{1+P(\boldsymbol{y}_{1:T})}{\gamma_i} + \frac{\gamma_i KT}{\beta^2\rho^2} + T\epsilon + \frac{1}{\epsilon}\right)$, *where $P(\boldsymbol{y}_{1:T}) = \sum_{t \in [T-1]}\|\boldsymbol{y}_t - \boldsymbol{y}_{t+1}\|$, and parameters $(\alpha, \gamma_i, \epsilon, \rho)$ are specified in Algorithm 1.*

*Proof sketch.* We define the smoothed-versions of BOCO rewards $h_t : \mathcal{X} \to \mathbb{R}$ as $\hat{h}_t(\boldsymbol{x}) = \mathbb{E}_{\boldsymbol{v} \sim U(\mathbb{B})}[\mathcal{L}_t(\boldsymbol{x} + \rho\boldsymbol{v}, \boldsymbol{\lambda}_t)]$. Then, we decompose the primal ascent loss (BOCO dynamic regret) $\sum_{\tau \in [t]}(h_\tau(\boldsymbol{y}_\tau) - h_\tau(\boldsymbol{x}_\tau))$ into 3 terms:

$$\sum_{\tau \in [t]}(h_\tau(\boldsymbol{y}_\tau) - \hat{h}_\tau((1-\alpha)\boldsymbol{y}_\tau)) + \sum_{\tau \in [t]}(\hat{h}_\tau((1-\alpha)\boldsymbol{y}_\tau) - \hat{h}_\tau(\widetilde{\boldsymbol{x}}_\tau)) + \sum_{\tau \in [t]}(\hat{h}_\tau(\widetilde{\boldsymbol{x}}_\tau) - h_\tau(\boldsymbol{x}_\tau)),$$

where we recall the $\widetilde{\boldsymbol{x}}_\tau$ is the un-perturbed version of our primal lever decision in Algorithm 1. The first and third summands can be directly be bounded via exploiting Lipschtiz continuity properties for $h_t$ (see Lemma A.1). The second summand can further be upper bounded (using Lemma A.2) by the surrogate loss difference between the primal decision weighted over all expert decisions and the comparator decision, namely $\ell_\tau(\widetilde{\boldsymbol{x}}_\tau) - \ell_\tau((1-\alpha)\boldsymbol{y}_\tau)$. Then, we choose any expert $i \in [N]$ corresponding to primal ascent step size $\gamma_i > 0$ as an "intermediary" and decompose cumulative surrogate loss as $\sum_{t \in [T]}(\ell_t(\widetilde{\boldsymbol{x}}_t) - \ell_t((1-\alpha)\boldsymbol{y}_t)) = \sum_{t \in [T]}(\ell_t(\widetilde{\boldsymbol{x}}_t) - \ell_t(\widetilde{\boldsymbol{x}}_t^i)) + \sum_{t \in [T]}(\ell_t(\widetilde{\boldsymbol{x}}_t^i) - \ell_t((1-\alpha)\boldsymbol{y}_t))$. The first summand corresponds to the cumulative loss of our unperturbed primal decisions against any expert, and the second summand represents the cumulative loss of any expert w.r.t. the comparator sequence. Both summands are bounded in Lemma A.3. $\square$

We note that choosing different experts as the intermediary may yield different bounds. But since Lemma A.3 holds for any expert $i \in [N]$, we can consider the expert with the "optimal" primal ascent step size in each world, respectively.

## 5 Additional discussions

**Lower bounds.** A natural question to ask is whether regret bounds in Theorem 4.2 are optimal. Here, we look at lower bounds in "relaxed" problem settings, e.g., bandit online convex optimization with no-constraints [11, 44], or online constrained optimization under full-information feedback [8, 10]:

| Stochastic | $\delta$-corrupted | Adversarial | Periodic | Ergodic |
|---|---|---|---|---|
| $\Omega(\sqrt{T})$ [19] | $\Omega(\sqrt{T} + \delta)$ [10] | $\Omega\left(\left(1 - \frac{1}{\xi}\right)T\right)$ [8] | $\Omega(\sqrt{qT})$[10] | NA |

These relaxed lower bounds are applicable to our setting (though they may not be the tightest bounds), and we can see that it may be possible to employ more complex machinery to attain better regret performances for our bandit convex online optimization setup with uncertain constraints. Yet, to identify an algorithm that can achieve the optimal regret in many worlds is an extremely challenging

open problem. For instance, even in the relaxed setting of bandit convex optimization with no-constraints, to the best of our knowledge there is no universal algorithm that achieves optimal regret in just the stochastic and adversarial worlds.

**Other applications.** Our proposed Algorithm 1 addresses a more general problem of online decision making with multi-dimensional decisions, bandit feedback and long-time uncertain constraints. That being said, our algorithm can also be applicable to other problem settings, for example:

*Personalized recommendation and assortment in online retailing.* Online retail platforms aim to optimize sales or revenue by recommending a limited assortment of products to customers. However, the likelihood of customer purchases given an assortment as well as their preferences are unknown, and different customer types may change over time due to evolving market trends. Platforms only observe bandit binary purchase decisions. Hence in our context, the decision set $\mathcal{X}$ represents a probability simplex over $d$ items. The online decisions $\boldsymbol{x}_t \in \mathcal{X}$ correspond to recommendation probabilities for each item. The function $f_t(\boldsymbol{x}_t)$ reflects the revenue generated from customer purchases, while $\boldsymbol{g}_t(\boldsymbol{x}_t)$ captures assortment capacity constraints, product inventory limits, or fairness considerations to ensure equal visibility for each product [18].

*Real time posted pricing in E-commerce and cloud computing.* In online E-commerce platforms such as Amazon and eBay, sellers set prices to sell various items to sequentially arriving customers [5, 34, 26]; and in cloud computing, operators of cloud services such as Alibaba Cloud (Alicloud) or Amazon Web Services (AWS) set prices for renting out different computing capacities (virtual machines, VMs etc.) upon customer requests [39]. The arrival of customer types as well as their demand may differ significantly over time (think about demand for sanitary products during pandemics outbursts, or computing resource demands during periodic business hours). Further, decision makers only get to observe (bandit) demand at the realized pricing decisions. Thereby, we can view $\boldsymbol{x}_t \in \mathcal{X}$ as the vector of prices for various sold products, $f_t(\boldsymbol{x}_t)$ as the generated revenue over a given period, $\boldsymbol{g}_t(\boldsymbol{x}_t)$ as real time product/resource capacity constraints, operating cost constraints, etc.

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

# Appendices for
# Online Ad Procurement in Non-stationary Autobidding Worlds

## A Proofs for Section 4

### A.1 Additional definitions for Section 4

**Definition A.1** (Total variation between probability distributions). *Consider two distributions* $\{\mathcal{P}, \mathcal{P}'\} \subseteq \Delta(\mathcal{S})$. *Then we define their total variation as* $\|\mathcal{P} - \mathcal{P}'\|_{TV} = \frac{1}{2}\int_{\mathcal{S}} |\mathcal{P}(s) - \mathcal{P}'(s)|ds$.

We also define the smoothed version of $h_t : \mathcal{X} \to \mathbb{R}$ (see Eq. (5)) for any $t$ as follows:

$$\hat{h}_t(\boldsymbol{x}) = \mathbb{E}_{\boldsymbol{v} \sim U(\mathbb{B})}[\mathcal{L}_t(\boldsymbol{x} + \rho\boldsymbol{v}, \boldsymbol{\lambda}_t)] \tag{11}$$

where we recall the Lagrangian function $\mathcal{L}_t$ is defined in Eq. (4).

### A.2 Additional lemmas for Section 4

**Lemma A.1** (Lipschitz continuity). *Let Assumption 2.1 hold, and we recall the definitions of $h_t(\boldsymbol{x})$ and $\hat{h}_t(\boldsymbol{x})$ from Eqs. (5) and (11), respectively, and recall $\boldsymbol{\lambda}_1 \ldots \boldsymbol{\lambda}_T$ as the dual variables generated from Algorithm 1. Then for any $\{\boldsymbol{x}, \boldsymbol{x}'\} \subseteq \mathcal{X}$, we have $|h_t(\boldsymbol{x}) - h_t(\boldsymbol{x}')| \leq (1 + K\frac{\bar{F}}{\beta})L \cdot \|\boldsymbol{x} - \boldsymbol{x}'\|$ and $\left|h_t(\boldsymbol{x}) - \hat{h}_t(\boldsymbol{x})\right| \leq (1 + K\frac{\bar{F}}{\beta})L\rho$.*

**Lemma A.2** (Bounding BOCO dynamic regret with surrogate loss). *Recall the definition $\hat{h}_t(\boldsymbol{x}) = \mathbb{E}_{\boldsymbol{v} \sim U(\mathbb{B})}[\mathcal{L}_t(\boldsymbol{x} + \rho\boldsymbol{v}, \boldsymbol{\lambda}_t)]$. Then, $\hat{h}_t(\boldsymbol{x})$ is concave. Further, For any $\boldsymbol{y} \in (1 - \alpha)\mathcal{X}$, we have $\hat{h}_t(\boldsymbol{y}) - \hat{h}_t(\widetilde{\boldsymbol{x}}_t) \leq \mathbb{E}_{\boldsymbol{u}_t \sim U(\mathbb{S})}[\ell_t(\widetilde{\boldsymbol{x}}_t) - \ell_t(\boldsymbol{y})]$, where $\widetilde{\boldsymbol{x}}_t$ is defined in Eq. (6), and the surrogate loss function $\ell_t : \mathcal{X} \to \mathbb{R}$ is defined in Eq. (8).*

**Lemma A.3** (Bounding surrogate loss for each expert). *Recall the definition of individual forecasters $\widetilde{\boldsymbol{x}}_t^i$ defined in Eq. (9), and the surrogate loss function $\ell_t : \mathcal{X} \to \mathbb{R}$ defined in Eq. (8). Then for any $i \in [N]$ and any sequence $\boldsymbol{y}_{1:T} \in \mathcal{X}^T$, by defining $P(\boldsymbol{y}_{1:T}) = \sum_{t \in [T-1]}\|\boldsymbol{y}_t - \boldsymbol{y}_{t+1}\|$ (see Lemma 4.6), we have (i) $\sum_{t \in [T]}(\ell_t(\widetilde{\boldsymbol{x}}_t^i) - \ell_t((1 - \alpha)\boldsymbol{y}_t)) \leq \mathcal{O}\left(\frac{1 + P(\boldsymbol{y}_{1:T})}{\gamma_i} + \frac{\gamma_i}{\beta^2\rho^2}T\right)$ and (ii) $\sum_{t \in [T]}(\ell_t(\widetilde{\boldsymbol{x}}_t) - \ell_t(\widetilde{\boldsymbol{x}}_t^i)) \leq \mathcal{O}(T\epsilon + \frac{1}{\epsilon})$. where the constant $\beta$ is specified in Algorithm 1. Here, recall $D$ is the diameter of the decision set $\mathcal{X}$.*

The proofs of Lemmas A.1, A.2, A.3 are shown in Appendices A.9, A.10, and A.11, respectively.

### A.3 Proof for Lemma 4.1

*Proof.* For any $k \in [K]$ we have

$$\sum_{t \in [T]} g_{k,t}(\boldsymbol{x}_t) = \sum_{t \in [\tau_A - 1]} g_{k,t}(\boldsymbol{x}_t) + \sum_{t=\tau_A}^{T} g_{k,t}(\boldsymbol{x}_t) \overset{(a)}{\geq} \sum_{t \in [\tau_A - 1]} g_{k,t}(\boldsymbol{x}_t) + \bar{\beta}(T - \tau_A + 1)$$

$$\geq \sum_{t \in [\tau_A - 1]} g_{k,t}(\boldsymbol{x}_t) + \beta(T - \tau_A) + \beta \overset{(b)}{\geq} \bar{G} + \beta > 0 , \tag{12}$$

where in $(a)$ we set $\boldsymbol{x}_t = \widetilde{\boldsymbol{x}}_\beta$ for all $t = \tau_A \ldots T$ and $g_{k,t}(\widetilde{\boldsymbol{x}}_\beta) \geq \bar{\beta}$ for any $k \in [K]$; $(b)$ follows from the definition of the stopping time such that for any $t' < \tau_A$ and $k \in [K]$ we have $\sum_{t \in [t']} g_{k,t}(\boldsymbol{x}_t) - \bar{G} + \beta(T - t' - 1) \geq 0$. $\square$

### A.4 Proof for Lemma 4.4

*Proof.* It is easy to see $\boldsymbol{\lambda}_{t+1} = \Pi_{[\boldsymbol{0}, \frac{\bar{F}}{\beta}\boldsymbol{e}]}(\boldsymbol{\lambda}_t - \eta\nabla_{\boldsymbol{\lambda}}\mathcal{L}_t(\boldsymbol{x}_t, \boldsymbol{\lambda}_t))_+ = \arg\min_{\boldsymbol{\lambda} \in [\boldsymbol{0}, \frac{\bar{F}}{\beta}\boldsymbol{e}]} \nabla_{\boldsymbol{\lambda}}\mathcal{L}_t(\boldsymbol{x}_t, \boldsymbol{\lambda}_t)^\top\boldsymbol{\lambda} + \frac{1}{2\eta}\|\boldsymbol{\lambda} - \boldsymbol{\lambda}_t\|^2$. By the first-order stationary condition

at $\boldsymbol{\lambda}_{t+1}$, we have for any $\boldsymbol{\lambda} \in [\mathbf{0}, \frac{\bar{F}}{\beta}\boldsymbol{e}]$

$$\left(\nabla_{\boldsymbol{\lambda}}\mathcal{L}_t(\boldsymbol{x}_t, \boldsymbol{\lambda}_t) + \frac{1}{\eta}(\boldsymbol{\lambda}_{t+1} - \boldsymbol{\lambda}_t)\right)^\top (\boldsymbol{\lambda} - \boldsymbol{\lambda}_{t+1}) \geq 0 .$$

Then for all $\boldsymbol{\lambda} \in \mathbb{R}_{\geq 0}^K$, it follows that

$$
\begin{aligned}
&\nabla_{\boldsymbol{\lambda}}\mathcal{L}_t(\boldsymbol{x}_t, \boldsymbol{\lambda}_t)^\top (\boldsymbol{\lambda}_t - \boldsymbol{\lambda}) \\
={}& \nabla_{\boldsymbol{\lambda}}\mathcal{L}_t(\boldsymbol{x}_t, \boldsymbol{\lambda}_t)^\top (\boldsymbol{\lambda}_t - \boldsymbol{\lambda}_{t+1}) + \nabla_{\boldsymbol{\lambda}}\mathcal{L}_t(\boldsymbol{x}_t, \boldsymbol{\lambda}_t)^\top (\boldsymbol{\lambda}_{t+1} - \boldsymbol{\lambda}) \\
\leq{}& \nabla_{\boldsymbol{\lambda}}\mathcal{L}_t(\boldsymbol{x}_t, \boldsymbol{\lambda}_t)^\top (\boldsymbol{\lambda}_t - \boldsymbol{\lambda}_{t+1}) + \frac{1}{\eta}(\boldsymbol{\lambda}_{t+1} - \boldsymbol{\lambda}_t)^\top (\boldsymbol{\lambda} - \boldsymbol{\lambda}_{t+1}) \\
\leq{}& \nabla_{\boldsymbol{\lambda}}\mathcal{L}_t(\boldsymbol{x}_t, \boldsymbol{\lambda}_t)^\top (\boldsymbol{\lambda}_t - \boldsymbol{\lambda}_{t+1}) + \frac{1}{2\eta}\|\boldsymbol{\lambda} - \boldsymbol{\lambda}_t\|^2 - \frac{1}{2\eta}\|\boldsymbol{\lambda} - \boldsymbol{\lambda}_{t+1}\|^2 - \frac{1}{2\eta}\|\boldsymbol{\lambda}_{t+1} - \boldsymbol{\lambda}_t\|^2 \\
\leq{}& \frac{\eta}{2}\|\nabla_{\boldsymbol{\lambda}}\mathcal{L}_t(\boldsymbol{x}_t, \boldsymbol{\lambda}_t)\|^2 + \frac{1}{2\eta}\|\boldsymbol{\lambda} - \boldsymbol{\lambda}_t\|^2 - \frac{1}{2\eta}\|\boldsymbol{\lambda} - \boldsymbol{\lambda}_{t+1}\|^2 .
\end{aligned}
$$

By a telescoping argument, we have

$$
\begin{aligned}
\sum_{\tau \in [t]} \nabla_{\boldsymbol{\lambda}}\mathcal{L}_\tau(\boldsymbol{x}_\tau, \boldsymbol{\lambda}_\tau)^\top (\boldsymbol{\lambda}_\tau - \boldsymbol{\lambda}) &\leq \frac{\eta}{2}\sum_{\tau \in [t]} \|\nabla_{\boldsymbol{\lambda}}\mathcal{L}_\tau(\boldsymbol{x}_\tau, \boldsymbol{\lambda}_\tau)\|^2 + \frac{1}{2\eta}\|\boldsymbol{\lambda} - \boldsymbol{\lambda}_1\|^2 \\
&= \frac{\eta}{2}\sum_{\tau \in [t]} \|\nabla_{\boldsymbol{\lambda}}\mathcal{L}_\tau(\boldsymbol{x}_\tau, \boldsymbol{\lambda}_\tau)\|^2 + \frac{1}{2\eta}\|\boldsymbol{\lambda}\|^2 ,
\end{aligned}
\tag{13}
$$

where in the final equality we used $\boldsymbol{\lambda}_1 = \mathbf{0}$. Also,

$$\|\nabla_{\boldsymbol{\lambda}}\mathcal{L}_\tau(\boldsymbol{x}_\tau, \boldsymbol{\lambda}_\tau)\|^2 = \|\boldsymbol{g}_\tau(\boldsymbol{x}_\tau)\|^2 \leq K\bar{G}^2 . \tag{14}$$

Hence, combining Eqs. (13) and (14), we get the desired bound. □

## A.5 Proof of Lemma 4.5

*Proof.* If $\tau_A = T$, taking $\boldsymbol{\lambda} = 0$ in Lemma 4.4 yields $\sum_{t \in [T]} \boldsymbol{\lambda}_t^\top \boldsymbol{g}_t(\boldsymbol{x}_t) \leq \frac{\eta}{2}TK\bar{G}^2$ and thus the desired inequality holds. If $\tau_A < T$, then there exists some $k \in [K]$ such that $\sum_{t \in [\tau_A]} g_{k,t}(\boldsymbol{x}_t) - \bar{G} + \beta(T - \tau_A - 1) < 0$, so by taking $\boldsymbol{\lambda} = \frac{\bar{F}}{\beta}\boldsymbol{e}_k$ ($\boldsymbol{e}_k \in \mathbb{R}^K$ is the unit vector whose $k$th entry is 1) in Lemma 4.4 yields

$$
\begin{aligned}
\sum_{t \in [\tau_A]} \boldsymbol{\lambda}_t^\top \boldsymbol{g}_t(\boldsymbol{x}_t) &\leq \sum_{t \in [\tau_A]} \boldsymbol{\lambda}^\top \boldsymbol{g}_t(\boldsymbol{x}_t) + \frac{\eta}{2}TK\bar{G}^2 + \frac{1}{2\eta}\|\boldsymbol{\lambda}\|^2 \\
&= \frac{\bar{F}}{\beta}\sum_{t \in [\tau_A]} g_{k,t}(\boldsymbol{x}_t) + \frac{\eta}{2}TK\bar{G}^2 + \frac{1}{2\eta}\left(\frac{\bar{F}}{\beta}\right)^2 \\
&\leq -\frac{\bar{F}}{\beta} \cdot \beta(T - \tau_A - 1) + \frac{\bar{F}}{\beta}\bar{G} + \frac{\eta}{2}TK\bar{G}^2 + \frac{1}{2\eta}\left(\frac{\bar{F}}{\beta}\right)^2 \\
&= -\bar{F}(T - \tau_A) + \bar{F} + \frac{\bar{F}}{\beta}\bar{G} + \frac{\eta}{2}TK\bar{G}^2 + \frac{1}{2\eta}\left(\frac{\bar{F}}{\beta}\right)^2 ,
\end{aligned}
$$

which completes the proof. □

## A.6 Proof of Lemma 4.6

*Proof.* Recall the definition of $h_t(\boldsymbol{x})$ in Eq. (5). Then, we have

$$
\begin{aligned}
&\sum_{\tau \in [t]} h_\tau(\boldsymbol{y}_\tau) - \sum_{\tau \in [t]} h_\tau(\boldsymbol{x}_\tau) \\
={}& \sum_{\tau \in [t]} \Big( \underbrace{h_\tau(\boldsymbol{y}_\tau) - \hat{h}_\tau((1-\alpha)\boldsymbol{y}_\tau)}_{A} + \underbrace{\hat{h}_\tau((1-\alpha)\boldsymbol{y}_\tau) - \hat{h}_\tau(\widetilde{\boldsymbol{x}}_\tau)}_{B} + \underbrace{\hat{h}_\tau(\widetilde{\boldsymbol{x}}_\tau) - h_\tau(\boldsymbol{x}_\tau)}_{C} \Big)
\end{aligned}
\tag{15}
$$

**Bounding $A$.**

$$
\begin{aligned}
h_\tau(\boldsymbol{y}_\tau) - \hat{h}_\tau((1-\alpha)\boldsymbol{y}_\tau) \;=\; & h_\tau(\boldsymbol{y}_\tau) - h_\tau((1-\alpha)\boldsymbol{y}_\tau) + h_\tau((1-\alpha)\boldsymbol{y}_\tau) - \hat{h}_\tau((1-\alpha)\boldsymbol{y}_\tau) \\
& \overset{(a)}{\leq} \; (1+K\frac{\bar{F}}{\beta})L\alpha\|\boldsymbol{y}_\tau\| + (1+K\frac{\bar{F}}{\beta})L\rho \\
& \overset{(b)}{\leq} \; (1+K\frac{\bar{F}}{\beta})L\alpha D + (1+K\frac{\bar{F}}{\beta})L\rho\,,
\end{aligned}
\tag{16}
$$

where (a) follows from Lemma A.1; (b) follows from $\|\boldsymbol{y}_\tau\| = \|\boldsymbol{y}_\tau - \boldsymbol{0}\| \leq D$ since we assumed $\boldsymbol{0} \in \mathcal{X}$.

**Bounding $B$.**

$$
\begin{aligned}
& \sum_{\tau \in [t]} \hat{h}_\tau((1-\alpha)\boldsymbol{y}_\tau) - \hat{h}_\tau(\widetilde{\boldsymbol{x}}_\tau) \\
& \overset{(a)}{\leq} \sum_{\tau \in [t]} \mathbb{E}_{\boldsymbol{u}_\tau \sim U(\mathbb{S})} \left[\ell_\tau(\widetilde{\boldsymbol{x}}_\tau) - \ell_\tau((1-\alpha)\boldsymbol{y}_\tau)\right] \\
& = \sum_{\tau \in [t]} \mathbb{E}_{\boldsymbol{u}_\tau \sim U(\mathbb{S})} \left[\ell_\tau(\widetilde{\boldsymbol{x}}_\tau) - \ell_\tau(\widetilde{\boldsymbol{x}}_\tau^i) + \ell_\tau(\widetilde{\boldsymbol{x}}_\tau^i) - \ell_\tau((1-\alpha)\boldsymbol{y}_\tau)\right] \\
& \overset{(b)}{\leq} \mathcal{O}\left(\frac{P(\boldsymbol{y}_{1:T})}{\gamma_i} + \frac{\gamma_i K \frac{\bar{F}}{\beta} T}{\rho^2} + T\epsilon + \frac{1}{\epsilon}\right),
\end{aligned}
\tag{17}
$$

where (a) follows from Lemma A.2 and (b) follows from Lemma A.3 (i) and (ii).

**Bounding $C$.**

$$
\begin{aligned}
\hat{h}_\tau(\widetilde{\boldsymbol{x}}_\tau) - h_\tau(\boldsymbol{x}_\tau) \;=\; & \hat{h}_\tau(\widetilde{\boldsymbol{x}}_\tau) - h_\tau(\widetilde{\boldsymbol{x}}_\tau) + h_\tau(\widetilde{\boldsymbol{x}}_\tau) - h_\tau(\boldsymbol{x}_\tau) \\
& \overset{(a)}{\leq} \; (1+K\frac{\bar{F}}{\beta})L\rho + (1+K\frac{\bar{F}}{\beta})L \cdot \|\widetilde{\boldsymbol{x}}_\tau - \boldsymbol{x}_\tau\| \\
& \overset{(b)}{=} \; (1+K\frac{\bar{F}}{\beta})L\rho + (1+K\frac{\bar{F}}{\beta})L \cdot \|\rho\boldsymbol{u}_\tau\| \\
& \leq \; 2\rho(1+K\frac{\bar{F}}{\beta})L\,,
\end{aligned}
\tag{18}
$$

where (a) follows Lemma A.1; (b) is from the definition $\boldsymbol{x}_\tau = \widetilde{\boldsymbol{x}}_\tau + \rho\boldsymbol{u}_\tau$ in Algorithm 1. $\qquad\square$

### A.7 Proof of Lemma 4.3

In this section, we provide upper bounds of the regret term under five different environments of input procedures: stochastic, adversarial, $\delta$-corrupted, periodic, and ergodic. The structure of the following proof might seem similar to [10]; see [10, Sections 3-5]. However, we note that the proof techniques are fundamentally different since our paper considers the bandit feedback environment while [10] makes sequential decisions after observations.

**Stochastic.**

*Proof.* In the stochastic regime, we have $\mathcal{P} = \mathcal{P}_1 = \cdots = \mathcal{P}_T$ for some $\mathcal{P}$, and therefore we can rewrite $\mathrm{OPT}(\mathcal{P}_{1:T})$ in Eq. (1) as followed

$$
\mathrm{OPT}(\mathcal{P}_{1:T}) = \max_{\boldsymbol{x}_{1:T} \in \mathcal{X}^T} \sum_{t \in [T]} F(\boldsymbol{x}_t) \quad \text{s.t.} \sum_{t \in [T]} \boldsymbol{G}(\boldsymbol{x}_t) \geq \boldsymbol{0}\,.
$$

where we defined $F(\boldsymbol{x}) = \mathbb{E}_{(f,\boldsymbol{g})\sim\mathcal{P}}[f(\boldsymbol{x})]$, and $\boldsymbol{G}(\boldsymbol{x}) = \mathbb{E}_{(f,\boldsymbol{g})\sim\mathcal{P}}[\boldsymbol{g}(\boldsymbol{x})]$ for any $\boldsymbol{x} \in \mathcal{X}$. Hence, for any $\boldsymbol{\lambda} \geq \mathbf{0}$ we have

$$
\begin{aligned}
\text{OPT}(\mathcal{P}_{1:T}) &= \frac{T-\tau_A}{T}\text{OPT}(\mathcal{P}_{1:T}) + \frac{\tau_A}{T}\text{OPT}(\mathcal{P}_{1:T}) \\
&\leq (T-\tau_A)\bar{F} + \frac{\tau_A}{T}\max_{\boldsymbol{x}_{1:T}\in\mathcal{X}^T}\sum_{t\in[T]}\left(F(\boldsymbol{x}_t) + \boldsymbol{\lambda}^\top\boldsymbol{G}(\boldsymbol{x}_t)\right) \\
&= (T-\tau_A)\bar{F} + \frac{\tau_A}{T}\max_{\boldsymbol{x}\in\mathcal{X}}\sum_{t\in[T]}\left(F(\boldsymbol{x}) + \boldsymbol{\lambda}^\top\boldsymbol{G}(\boldsymbol{x})\right) \\
&= (T-\tau_A)\bar{F} + \tau_A\max_{\boldsymbol{x}\in\mathcal{X}}\left(F(\boldsymbol{x}) + \boldsymbol{\lambda}^\top\boldsymbol{G}(\boldsymbol{x})\right) ,
\end{aligned}
\tag{19}
$$

where in the inequality we applied Assumption 2.1 which states $\sup_{\boldsymbol{x}\in\mathcal{X}}|f(\boldsymbol{x})| \leq \bar{F}$ for all $(f,\boldsymbol{g}) \in \mathcal{S}$. Choosing $\boldsymbol{\lambda} = \bar{\boldsymbol{\lambda}}_{\tau_A} := \frac{1}{\tau_A}\sum_{t\in[\tau_A]}\boldsymbol{\lambda}_t$ we have

$$
\begin{aligned}
\text{OPT}(\mathcal{P}_{1:T}) &\leq \mathbb{E}\left[(T-\tau_A)\bar{F} + \tau_A\max_{\boldsymbol{x}\in\mathcal{X}}\left(F(\boldsymbol{x}) + \boldsymbol{\lambda}^\top\boldsymbol{G}(\boldsymbol{x})\right)\right] \\
&\leq \mathbb{E}\left[(T-\tau_A)\bar{F} + \max_{\boldsymbol{x}\in\mathcal{X}}\sum_{t\in[\tau_A]}(F(\boldsymbol{x}) + \boldsymbol{\lambda}_t^\top\boldsymbol{G}(\boldsymbol{x}))\right] \\
&\stackrel{(a)}{\leq} \mathbb{E}\left[(T-\tau_A)\bar{F} + \max_{\boldsymbol{x}\in\mathcal{X}}\sum_{t\in[\tau_A]}\mathbb{E}\left[f_t(\boldsymbol{x}) + \boldsymbol{\lambda}_t^\top\boldsymbol{g}_t(\boldsymbol{x})\,\Big|\,\sigma(\mathcal{H}_{t-1})\right]\right] \\
&\stackrel{(b)}{=} \mathbb{E}\left[(T-\tau_A)\bar{F} + \max_{\boldsymbol{x}\in\mathcal{X}}\sum_{t\in[\tau_A]}\mathbb{E}\left[h_t(\boldsymbol{x})\,\Big|\,\sigma(\mathcal{H}_{t-1})\right]\right] \\
&\leq \mathbb{E}\left[(T-\tau_A)\bar{F} + \max_{\boldsymbol{x}\in\mathcal{X}}\sum_{t\in[\tau_A]}h_t(\boldsymbol{x})\right] ,
\end{aligned}
\tag{20}
$$

where in $(a)$ we used the fact that $\boldsymbol{\lambda}_t$ is $\mathcal{H}_{t-1}$-measurable; in $(b)$ we used definitions $h_t(\boldsymbol{x}) = \mathcal{L}_t(\boldsymbol{x};\boldsymbol{\lambda}_t)$ and $\mathcal{L}_t(\boldsymbol{x};\boldsymbol{\lambda}) = f_t(\boldsymbol{x}) + \boldsymbol{\lambda}^\top\boldsymbol{g}_t(\boldsymbol{x})$ in Eqs. (4) and (5) respectively.

On the other hand, we have

$$
f_t(\boldsymbol{x}_t) = h_t(\boldsymbol{x}_t) - \boldsymbol{\lambda}_t^\top\boldsymbol{g}_t(\boldsymbol{x}_t),
\tag{21}
$$

so combining this with Eq. (20) we have

$$
\text{OPT}(\mathcal{P}_{1:T}) - \sum_{t\in[T]}\mathbb{E}[f_t(\boldsymbol{x}_t)] \leq \mathbb{E}\left[(T-\tau_A)\bar{F} + \max_{\boldsymbol{x}\in\mathcal{X}}\sum_{t\in[\tau_A]}\left(h_t(\boldsymbol{x}) - h_t(\boldsymbol{x}_t)\right) + \sum_{t\in\tau_A}\boldsymbol{\lambda}_t^\top\boldsymbol{g}_t(\boldsymbol{x}_t)\right]
\tag{22}
$$

where we also used the fact that $f_t(\boldsymbol{x}) \geq 0$ for all $t = \tau_A + 1 \ldots T$ and $\boldsymbol{x} \in \mathcal{X}$. $\qquad\square$

**Adversarial.**

*Proof.* Recall the definition of $\xi$ is Theorem 4.2:

$$
\xi = 1 - \frac{\min_{(f,\boldsymbol{g})\in\mathcal{S}}\min_{k\in[K],\boldsymbol{x}\in\mathcal{X}}g_k(\boldsymbol{x})}{\bar{\beta}} > 1 .
\tag{23}
$$

For any $t \in [T]$, define $\widetilde{\boldsymbol{y}}_t = \arg\max_{\boldsymbol{x}} f_t(\boldsymbol{x}) + \boldsymbol{\lambda}_t^\top\boldsymbol{g}_t(\boldsymbol{x})$.

By comparing to the safety action $\boldsymbol{x}_\beta \in \mathcal{X}$ which ensures $g_k(\boldsymbol{x}_\beta) \geq \bar{\beta}$ for any $k \in [K]$ and $(f,\boldsymbol{g}) \in \mathcal{S}$, as well as the optimal hindsight action $\boldsymbol{x}_t^* \in \mathcal{X}$ (i.e., $\boldsymbol{x}_1^* \ldots \boldsymbol{x}_T^*$ is the optimal decision sequence to $\text{OPT}(\mathcal{P}_{1:T})$), we have

$$
\begin{aligned}
f_t(\widetilde{\boldsymbol{y}}_t) + \boldsymbol{\lambda}_t^\top\boldsymbol{g}_t(\widetilde{\boldsymbol{y}}_t) &\geq f_t(\boldsymbol{x}_\beta) + \boldsymbol{\lambda}_t^\top\boldsymbol{g}_t(\boldsymbol{x}_\beta) \geq \bar{\beta}\boldsymbol{\lambda}_t^\top\boldsymbol{e} \\
\text{and}\quad f_t(\widetilde{\boldsymbol{y}}_t) + \boldsymbol{\lambda}_t^\top\boldsymbol{g}_t(\widetilde{\boldsymbol{y}}_t) &\geq f_t(\boldsymbol{x}_t^*) + \boldsymbol{\lambda}_t^\top\boldsymbol{g}_t(\boldsymbol{x}_t^*).
\end{aligned}
\tag{24}
$$

We further have

$$
\begin{aligned}
\xi f_t(\widetilde{\boldsymbol{y}}_t) &= f_t(\widetilde{\boldsymbol{y}}_t) + (\xi - 1)f_t(\widetilde{\boldsymbol{y}}_t) \\
&\overset{(a)}{\geq} f_t(\boldsymbol{x}_t^*) + \boldsymbol{\lambda}_t^\top \boldsymbol{g}_t(\boldsymbol{x}_t^*) - \boldsymbol{\lambda}_t^\top \boldsymbol{g}_t(\widetilde{\boldsymbol{y}}_t) + (\xi - 1)\left(-\boldsymbol{\lambda}_t^\top \boldsymbol{g}_t(\widetilde{\boldsymbol{y}}_t) + \bar{\beta}\boldsymbol{\lambda}_t^\top \boldsymbol{e}\right) \\
&= f_t(\boldsymbol{x}_t^*) + \boldsymbol{\lambda}_t^\top \boldsymbol{g}_t(\boldsymbol{x}_t^*) - \xi\boldsymbol{\lambda}_t^\top \boldsymbol{g}_t(\widetilde{\boldsymbol{y}}_t) + (\xi-1)\bar{\beta}\boldsymbol{\lambda}_t^\top \boldsymbol{e} \\
&\overset{(b)}{\geq} f_t(\boldsymbol{x}_t^*) - \xi\boldsymbol{\lambda}_t^\top \boldsymbol{g}_t(\widetilde{\boldsymbol{y}}_t)
\end{aligned}
\tag{25}
$$

where (a) follows Eq. (24); in (b) we used the fact that $g_{k,t}(\boldsymbol{x}_t^*) + (\xi - 1)\bar{\beta} \geq 0$ since we have $\min_{(f,g)\in\mathcal{S}} \min_{k\in[K],\boldsymbol{x}\in\mathcal{X}}(g_{k,t}(\boldsymbol{x}) + (\xi-1)\bar{\beta}) \geq 0$ (see Eq. (23)). Hence we have

$$
\begin{aligned}
&\text{OPT}(\mathcal{P}_{1:T}) - \sum_{t\in[T]} \mathbb{E}[f_t(\boldsymbol{x}_t)] \\
&= \left(1 - \frac{1}{\xi}\right)\text{OPT}(\mathcal{P}_{1:T}) + \sum_{t\in[T]} \mathbb{E}\left[\frac{1}{\xi}f_t(\boldsymbol{x}_t^*) - f_t(\boldsymbol{x}_t)\right] \\
&\leq \left(1 - \frac{1}{\xi}\right)\text{OPT}(\mathcal{P}_{1:T}) + \sum_{t\in[T]} \mathbb{E}\left[f_t(\widetilde{\boldsymbol{y}}_t) - f_t(\boldsymbol{x}_t) + \boldsymbol{\lambda}_t^\top \boldsymbol{g}_t(\widetilde{\boldsymbol{y}}_t)\right] \\
&\leq \left(1 - \frac{1}{\xi}\right)\text{OPT}(\mathcal{P}_{1:T}) + \mathbb{E}\left[(T - \tau_A)\bar{F} + \sum_{t\in\tau_A}\left(f_t(\widetilde{\boldsymbol{y}}_t) - f_t(\boldsymbol{x}_t) + \boldsymbol{\lambda}_t^\top \boldsymbol{g}_t(\widetilde{\boldsymbol{y}}_t)\right)\right],
\end{aligned}
\tag{26}
$$

which completes the proof. $\qquad\square$

**$\delta$-corrupted.**

Here, we will prove a more general $\delta$-corrupted model where the input distribution sequence $\mathcal{P}_{1:T}$ satisfies the following:

$$
\sum_{t\in[T]} \left\|\mathcal{P}_t - \frac{1}{T}\sum_{s\in[T]}\mathcal{P}_s\right\|_{TV} \leq \delta,
\tag{27}
$$

where the total variation norm is defined in Definition A.1. In fact, the definition in Section 2.2 for the $\delta$-corrupted regime satisfies the above property: recall the definition in Section 2.2, there exists $\mathcal{P} \in \Delta(\mathcal{S})$ as well as $\delta \in \mathbb{N}$ periods $\mathcal{T} = \{\tau_1 \dots \tau_\delta\} \subset [T]$ such that $\mathcal{P}_t = \mathcal{P}$ for all $t \notin \mathcal{T}$, so for any $t \notin \mathcal{T}$, we have

$$
\begin{aligned}
\left\|\mathcal{P}_t - \frac{1}{T}\sum_{s\in[T]}\mathcal{P}_s\right\|_{TV} &= \left\|\mathcal{P} - \frac{1}{T}\left(T\mathcal{P} + \sum_{s\in\mathcal{T}}(\mathcal{P}_s - \mathcal{P})\right)\right\|_{TV} \\
&= \left\|\frac{1}{T}\sum_{s\in\mathcal{T}}(\mathcal{P} - \mathcal{P}_s)\right\|_{TV} \leq \frac{\delta}{2T}.
\end{aligned}
\tag{28}
$$

On the other hand, we have for any $t \in \mathcal{T}$, $\left\|\mathcal{P}_t - \frac{1}{T}\sum_{s\in[T]}\mathcal{P}_s\right\|_{TV} \leq \frac{1}{2}$. After summing it up, we get

$$
\begin{aligned}
\sum_{t\in[T]} \left\|\mathcal{P}_t - \frac{1}{T}\sum_{s\in[T]}\mathcal{P}_s\right\|_{TV} &= \sum_{t\in\mathcal{T}} \left\|\mathcal{P}_t - \frac{1}{T}\sum_{s\in[T]}\mathcal{P}_s\right\|_{TV} + \sum_{t\notin\mathcal{T}} \left\|\mathcal{P}_t - \frac{1}{T}\sum_{s\in[T]}\mathcal{P}_s\right\|_{TV} \\
&\leq \frac{\delta}{2} + (T - \delta)\frac{\delta}{2T} \leq \delta,
\end{aligned}
$$

which coincides with our general definition of $\delta$-corruption in Eq. (27).

We now prove the $\delta$-corruption regime under the general definition in Eq. (27). Define $\widetilde{\mathcal{P}} = \frac{1}{T}\sum_{s\in[T]}\mathcal{P}_s$, $\widetilde{F}(\boldsymbol{x}) = \mathbb{E}_{(f,g)\sim\widetilde{\mathcal{P}}}[f(\boldsymbol{x})]$, $\widetilde{\boldsymbol{G}}(\boldsymbol{x}) = \mathbb{E}_{(f,g)\sim\widetilde{\mathcal{P}}}[\boldsymbol{g}(\boldsymbol{x})]$, $F_t(\boldsymbol{x}) = \mathbb{E}_{(f,g)\sim\mathcal{P}_t}[f(\boldsymbol{x})]$ and $\boldsymbol{G}_t(\boldsymbol{x}) = \mathbb{E}_{(f,g)\sim\mathcal{P}_t}[\boldsymbol{g}(\boldsymbol{x})]$ for all $t \in [T]$ and any $x \in \mathcal{X}$. Then for any $\boldsymbol{\lambda} \in [\boldsymbol{0}, \frac{\bar{F}}{\bar{\beta}}\boldsymbol{e}]$, we have

$$\text{OPT}(\mathcal{P}_{1:T}) \leq \max_{\boldsymbol{x}_{1:T} \in \mathcal{X}^T} \sum_{t \in [T]} \left( F_t(\boldsymbol{x}_t) + \boldsymbol{\lambda}^\top \boldsymbol{G}_t(\boldsymbol{x}_t) \right)$$

$$\leq \max_{\boldsymbol{x}_{1:T} \in \mathcal{X}^T} \sum_{t \in [T]} (\widetilde{F}(\boldsymbol{x}_t) + \boldsymbol{\lambda}^\top \widetilde{\boldsymbol{G}}(\boldsymbol{x}_t)) + (\bar{F} + \bar{G}K\frac{\bar{F}}{\beta})\delta \qquad (29)$$

$$= T \cdot \max_{\boldsymbol{x} \in \mathcal{X}} (\widetilde{F}(\boldsymbol{x}) + \boldsymbol{\lambda}^\top \widetilde{\boldsymbol{G}}(\boldsymbol{x})) + (\bar{F} + \bar{G}K\frac{\bar{F}}{\beta})\delta,$$

where the last inequality follows the definitions of $(\widetilde{F}, \widetilde{\boldsymbol{G}})$, Assumption 2.1, and the general definition of $\delta$-corruption in Eq. (27). After choosing $\bar{\boldsymbol{\lambda}} = \frac{1}{\tau_A}\sum_{t \in [\tau_A]} \boldsymbol{\lambda}_t$, similar to our proof in Eq. (20) for the stochastic case we have

$$\text{OPT}(\mathcal{P}_{1:T})$$

$$= \mathbb{E}\Big[\frac{T - \tau_A}{T}\text{OPT}(\mathcal{P}_{1:T}) + \frac{\tau_A}{T}\text{OPT}(\mathcal{P}_{1:T})\Big]$$

$$\overset{(a)}{\leq} \mathbb{E}\Big[(T - \tau_A)\bar{F} + \tau_A \cdot \max_{\boldsymbol{x} \in \mathcal{X}}(\widetilde{F}(\boldsymbol{x}) + \boldsymbol{\lambda}^\top \widetilde{\boldsymbol{G}}(\boldsymbol{x})) + \frac{\tau_A}{T}(\bar{F} + \bar{G}K\frac{\bar{F}}{\beta})\delta\Big]$$

$$= \mathbb{E}\Big[(T - \tau_A)\bar{F} + \frac{\tau_A}{T}(\bar{F} + \bar{G}K\frac{\bar{F}}{\beta})\delta + \max_{\boldsymbol{x} \in \mathcal{X}}\Big( \sum_{t \in [\tau_A]} \widetilde{F}(\boldsymbol{x}) + \boldsymbol{\lambda}_t^\top \widetilde{\boldsymbol{G}}(\boldsymbol{x})\Big)\Big]$$

$$\overset{(b)}{\leq} \mathbb{E}\Big[(T - \tau_A)\bar{F} + \Big(1 + \frac{\tau_A}{T}\Big)(\bar{F} + \bar{G}K\frac{\bar{F}}{\beta})\delta + \max_{\boldsymbol{x} \in \mathcal{X}} \sum_{t \in [\tau_A]} (F_t(\boldsymbol{x}) + \boldsymbol{\lambda}_t^\top \boldsymbol{G}_t(\boldsymbol{x}))\Big]$$

$$\leq \mathbb{E}\Big[(T - \tau_A)\bar{F} + \Big(1 + \frac{\tau_A}{T}\Big)(\bar{F} + \bar{G}K\frac{\bar{F}}{\beta})\delta + \max_{\boldsymbol{x} \in \mathcal{X}} \sum_{t \in [\tau_A]} \mathbb{E}\Big[f_t(\boldsymbol{x}) + \boldsymbol{\lambda}_t^\top \boldsymbol{g}_t(\boldsymbol{x}) \,\Big|\, \sigma(\mathcal{H}_{t-1})\Big]\Big]$$

$$\leq \mathbb{E}\Big[(T - \tau_A)\bar{F} + 2(\bar{F} + \bar{G}K\frac{\bar{F}}{\beta})\delta + \max_{\boldsymbol{x} \in \mathcal{X}} \sum_{t \in [\tau_A]} h_t(\boldsymbol{x})\Big],$$

$$(30)$$

where (a) follows from Eq. (29); (b) follows from the definition of general $\delta$-corruption in Eq. (27).

Finally, we complete the proof by using the definition $f_t(\boldsymbol{x}_t) = h_t(\boldsymbol{x}_t) - \boldsymbol{\lambda}_t^\top \boldsymbol{g}_t(\boldsymbol{x}_t)$ and following the same argument as in Eq. (22) for the stochastic regime.

**Periodic.**

Recall in Section 2.2 that in the periodic regime, there exists cycle length $q \in \mathbb{N}$ such that $T = cq$ for some integer $c \geq 2$ with $\mathcal{P}_{1:T}$ as $\mathcal{P}_{1:q} = \mathcal{P}_{q+1:2q} = \cdots = \mathcal{P}_{(c-1)q+1:T}$. For any $t \in [T]$, define $c_t \in [c]$ such that $(c_t - 1)q + 1 \leq t \leq c_t q$. After denoting $\widetilde{\mathcal{P}} = \frac{1}{q}\sum_{t \in [q]} \mathcal{P}_t$, we define the mean deviation within a single cycle of length $q$ as

$$MD(\mathcal{P}_{1:q}) = \sum_{1 \leq t \leq q} \|\mathcal{P}_t - \widetilde{\mathcal{P}}\|_{TV} \quad \text{and} \quad \delta = c \cdot MD(\mathcal{P}_{1:q}). \qquad (31)$$

We define $\widetilde{F}(\boldsymbol{x}) = \mathbb{E}_{(f,\boldsymbol{g}) \sim \widetilde{\mathcal{P}}}[f(\boldsymbol{x})]$, $\widetilde{\boldsymbol{G}}(\boldsymbol{x}) = \mathbb{E}_{(f,\boldsymbol{g}) \sim \widetilde{\mathcal{P}}}[\boldsymbol{g}(\boldsymbol{x})]$, $F_t(\boldsymbol{x}) = \mathbb{E}_{(f,\boldsymbol{g}) \sim \mathcal{P}_t}[f(\boldsymbol{x})]$ and $\boldsymbol{G}_t(\boldsymbol{x}) = \mathbb{E}_{(f,\boldsymbol{g}) \sim \mathcal{P}_t}[\boldsymbol{g}(\boldsymbol{x})]$ for all $t \in [T]$ and any $x \in \mathcal{X}$. Then for any $\boldsymbol{\lambda} \in [\boldsymbol{0}, \frac{\bar{F}}{\beta}\boldsymbol{e}]$, we have

$$\text{OPT}(\mathcal{P}_{1:T}) \leq \max_{\boldsymbol{x}_{1:T} \in \mathcal{X}^T} \sum_{t \in [T]} \left( F_t(\boldsymbol{x}_t) + \boldsymbol{\lambda}^\top \boldsymbol{G}_t(\boldsymbol{x}_t) \right)$$

$$= c \cdot \max_{\boldsymbol{x}_{1:q} \in \mathcal{X}^q} \sum_{t \in [q]} \left( F_t(\boldsymbol{x}_t) + \boldsymbol{\lambda}^\top \boldsymbol{G}_t(\boldsymbol{x}_t) \right)$$

$$\leq cq \cdot \max_{\boldsymbol{x} \in \mathcal{X}} (\widetilde{F}(\boldsymbol{x}) + \boldsymbol{\lambda}^\top \widetilde{\boldsymbol{G}}(\boldsymbol{x})) + (\bar{F} + \bar{G}K\frac{\bar{F}}{\beta})c \cdot MD(\mathcal{P}_{1:q})$$

$$\leq cq \cdot \max_{\boldsymbol{x} \in \mathcal{X}} (\widetilde{F}(\boldsymbol{x}) + \boldsymbol{\lambda}^\top \widetilde{\boldsymbol{G}}(\boldsymbol{x})) + (\bar{F} + \bar{G}K\frac{\bar{F}}{\beta})\delta,$$

where the equality follows the nature of periodic setting and the last inequality follows the definitions of $(\widetilde{F}, \widetilde{G})$, Assumption 2.1, and (31). After choosing $\boldsymbol{\lambda} = \sum_{\hat{c} \in [c_{\tau_A} - 1]} \frac{q}{\tau_A} \boldsymbol{\lambda}_{(\hat{c}-1)q+1} + \frac{\tau_A - (c_{\tau_A} - 1)q}{\tau_A} \boldsymbol{\lambda}_{(c_{\tau_A} - 1)q+1}$, we further have that

$$\text{OPT}(\mathcal{P}_{1:T})$$

$$= \frac{T - \tau_A}{T} \text{OPT}(\mathcal{P}_{1:T}) + \frac{\tau_A}{T} \text{OPT}(\mathcal{P}_{1:T})$$

$$\leq (T - \tau_A)\bar{F} + \tau_A \cdot \max_{\boldsymbol{x} \in \mathcal{X}}(\widetilde{F}(\boldsymbol{x}) + \boldsymbol{\lambda}^\top \widetilde{G}(\boldsymbol{x})) + \frac{\tau_A}{T}(\bar{F} + \bar{G}K\frac{\bar{F}}{\beta})\delta$$

$$= (T - \tau_A)\bar{F} + \max_{\boldsymbol{x} \in \mathcal{X}} \left( \tau_A \widetilde{F}(\boldsymbol{x}) + \left( \sum_{\hat{c} \in [c_{\tau_A} - 1]} q\boldsymbol{\lambda}_{(\hat{c}-1)q+1} + (\tau_A - (c_{\tau_A} - 1)q)\boldsymbol{\lambda}_{(c_{\tau_A} - 1)q+1} \right)^\top \widetilde{G}(\boldsymbol{x}) \right)$$

$$+ \frac{\tau_A}{T}(\bar{F} + \bar{G}K\frac{\bar{F}}{\beta})\delta$$

$$= (T - \tau_A)\bar{F} + \max_{\boldsymbol{x} \in \mathcal{X}} \left( q \cdot \sum_{\hat{c} \in [c_{\tau_A} - 1]} \left( \widetilde{F}(\boldsymbol{x}) + \boldsymbol{\lambda}_{(\hat{c}-1)q+1}^\top \widetilde{G}(\boldsymbol{x}) \right) \right.$$

$$+ (\tau_A - (c_{\tau_A} - 1)q) \cdot \left( \widetilde{F}(\boldsymbol{x}) + \boldsymbol{\lambda}_{(c_{\tau_A} - 1)q+1}^\top \widetilde{G}(\boldsymbol{x}) \right) \bigg) + \frac{\tau_A}{T}(\bar{F} + \bar{G}K\frac{\bar{F}}{\beta})\delta$$

$$\leq (T - \tau_A)\bar{F} + \max_{\boldsymbol{x} \in \mathcal{X}} \sum_{t \in [\tau_A]} \left( \widetilde{F}(\boldsymbol{x}) + \boldsymbol{\lambda}_t^\top \widetilde{G}(\boldsymbol{x}) \right) + \bar{G} \cdot \sum_{t \in [\tau_A]} \|\boldsymbol{\lambda}_t - \boldsymbol{\lambda}_{(c_t - 1)q+1}\|_1 + \frac{\tau_A}{T}(\bar{F} + \bar{G}K\frac{\bar{F}}{\beta})\delta.$$

From Eq. (9) in Algorithm 1, we know that $\|\boldsymbol{\lambda}_{t+1} - \boldsymbol{\lambda}_t\|_1 \leq \eta\bar{G}K$, which further implies $\|\boldsymbol{\lambda}_{t+i} - \boldsymbol{\lambda}_t\|_1 \leq \eta\bar{G}Ki$ for any $i \in [q-1]$ and thus

$$\sum_{t \in [\tau_A]} \|\boldsymbol{\lambda}_t - \boldsymbol{\lambda}_{(c_t - 1)q+1}\|_1 \leq c_{\tau_A} \eta\bar{G}K \sum_{i \in [q-1]} i \leq \frac{1}{2}\bar{G}K\eta c_{\tau_A} q^2. \tag{32}$$

After combining the two equations above, it follows that

$$\text{OPT}(\mathcal{P}_{1:T})$$

$$\leq \mathbb{E}\left[ (T - \tau_A)\bar{F} + \max_{\boldsymbol{x} \in \mathcal{X}} \sum_{t \in [\tau_A]} \left( \widetilde{F}(\boldsymbol{x}) + \boldsymbol{\lambda}_t^\top \widetilde{G}(\boldsymbol{x}) \right) + \frac{1}{2}\bar{G}^2 K\eta c_{\tau_A} q^2 + \frac{\tau_A}{T}(\bar{F} + \bar{G}K\frac{\bar{F}}{\beta})\delta \right]$$

$$\leq \mathbb{E}\left[ (T - \tau_A)\bar{F} + \max_{\boldsymbol{x} \in \mathcal{X}} \sum_{t \in [\tau_A]} \left( F_t(\boldsymbol{x}) + \boldsymbol{\lambda}_t^\top \boldsymbol{G}_t(\boldsymbol{x}) \right) + \frac{1}{2}\bar{G}^2 K\eta c_{\tau_A} q^2 + 2(\bar{F} + \bar{G}K\frac{\bar{F}}{\beta})\delta \right]$$

$$\leq \mathbb{E}\left[ (T - \tau_A)\bar{F} + 2(\bar{F} + \bar{G}K\frac{\bar{F}}{\beta})\delta + \frac{1}{2}\bar{G}^2 K\eta qT + \max_{\boldsymbol{x} \in \mathcal{X}} \sum_{t \in [\tau_A]} \mathbb{E}\left[ h_t(\boldsymbol{x}) \mid \sigma(\mathcal{H}_{t-1}) \right] \right]$$

$$\leq \mathbb{E}\left[ (T - \tau_A)\bar{F} + 2(\bar{F} + \bar{G}K\frac{\bar{F}}{\beta})\delta + \frac{1}{2}\bar{G}^2 K\eta qT + \max_{\boldsymbol{x} \in \mathcal{X}} \sum_{t \in [\tau_A]} h_t(\boldsymbol{x}) \right],$$

where the second last inequality follows from $c_{\tau_A} q \leq cq = T$.

Finally, we complete the proof by using the definition $f_t(\boldsymbol{x}_t) = h_t(\boldsymbol{x}_t) - \boldsymbol{\lambda}_t^\top \boldsymbol{g}_t(\boldsymbol{x}_t)$ and following the same argument as in Eq. (22) for the stochastic regime.

**Ergodic.**

Consider some $\kappa \geq \log(T)$. Given the input distribution sequence $\mathcal{P}_{1:T}$, denote $\mathcal{P}_{(t+\kappa)|[t-1]}$ as the conditional distribution of $(f_{t+\kappa}, g_{t+\kappa})$ conditioned on the $\{(f_\tau, g_\tau)\}_{\tau \in [t]}$. Then, in the ergodic regime, there exists a stationary distribution $\widetilde{\mathcal{P}} \in \Delta(\mathcal{S})$ and absolute constant $R > 0$ such that

$$\sup_{\{(f_t, g_t)\}_{t \in [T]} \in \mathcal{S}^T} \sup_{t \in [T - \kappa]} \|\mathcal{P}_{(t+\kappa)|[t-1]} - \widetilde{\mathcal{P}}\|_{TV} \leq \delta := R\exp(-\kappa); \tag{33}$$

see [10, Section 5.2] for further discussions. By defining $\widetilde{F}(\boldsymbol{x}) = \mathbb{E}_{(f,g)\sim\widetilde{\mathcal{P}}}[f(\boldsymbol{x})]$, $\widetilde{\boldsymbol{G}}(\boldsymbol{x}) = \mathbb{E}_{(f,g)\sim\widetilde{\mathcal{P}}}[\boldsymbol{g}(\boldsymbol{x})]$, $\hat{F}_{t+\kappa}(\boldsymbol{x}) = \mathbb{E}_{(f,g)\sim\mathcal{P}_{(t+\kappa)|[t-1]}}[f(\boldsymbol{x})]$, $\hat{\boldsymbol{G}}_{t+\kappa}(\boldsymbol{x}) = \mathbb{E}_{(f,g)\sim\mathcal{P}_{(t+\kappa)|[t-1]}}[\boldsymbol{g}(\boldsymbol{x})]$, $F_t(\boldsymbol{x}) = \mathbb{E}_{(f,g)\sim\mathcal{P}_t}[f(\boldsymbol{x})]$ and $\boldsymbol{G}_t(\boldsymbol{x}) = \mathbb{E}_{(f,g)\sim\mathcal{P}_t}[\boldsymbol{g}(\boldsymbol{x})]$ for all $t \in [T]$ and any $x \in \mathcal{X}$, we know that for any $\boldsymbol{\lambda} \in [\boldsymbol{0}, \frac{\bar{F}}{\beta}\boldsymbol{e}]$, it follows that

$$
\begin{aligned}
&\text{OPT}(\mathcal{P}_{1:T}) \\
&\leq \max_{\boldsymbol{x}_{1:T}\in\mathcal{X}^T} \mathbb{E}\left[\sum_{t\in[T]}\left(F_t(\boldsymbol{x}_t) + \boldsymbol{\lambda}^\top\boldsymbol{G}_t(\boldsymbol{x}_t)\right)\right] \\
&= \max_{\boldsymbol{x}_{1:\kappa}\in\mathcal{X}^\kappa} \mathbb{E}\left[\sum_{t\in[\kappa]}\left(F_t(\boldsymbol{x}_t) + \boldsymbol{\lambda}^\top\boldsymbol{G}_t(\boldsymbol{x}_t)\right)\right] + \max_{\boldsymbol{x}_{\kappa+1:T}\in\mathcal{X}^{T-\kappa}} \mathbb{E}\left[\sum_{t=1}^{T-\kappa}(\hat{F}_{t+\kappa}(\boldsymbol{x}_{t+\kappa}) + \boldsymbol{\lambda}^\top\hat{\boldsymbol{G}}_{t+\kappa}(\boldsymbol{x}_{t+\kappa}))\right] \\
&\leq (\bar{F} + \bar{G}K\frac{\bar{F}}{\beta})\kappa + \max_{\boldsymbol{x}_{\kappa+1:T}\in\mathcal{X}^{T-\kappa}} \sum_{t=1}^{T-\kappa}(\widetilde{F}(\boldsymbol{x}_{t+\kappa}) + \boldsymbol{\lambda}^\top\widetilde{\boldsymbol{G}}(\boldsymbol{x}_{t+\kappa})) + (\bar{F} + \bar{G}K\frac{\bar{F}}{\beta})\cdot(T-\kappa)\delta \\
&\leq T\cdot\max_{\boldsymbol{x}\in\mathcal{X}}(\widetilde{F}(\boldsymbol{x}) + \boldsymbol{\lambda}^\top\widetilde{\boldsymbol{G}}(\boldsymbol{x})) + (\bar{F} + \bar{G}K\frac{\bar{F}}{\beta})\kappa + (\bar{F} + \bar{G}K\frac{\bar{F}}{\beta})\cdot T\delta\,.
\end{aligned}
$$

$$(34)$$

By choosing $\boldsymbol{\lambda} = \frac{1}{\tau_A}\sum_{t\in[\tau_A]}\boldsymbol{\lambda}_t$, we further have

$$
\begin{aligned}
&\text{OPT}(\mathcal{P}_{1:T}) \\
&= \mathbb{E}\left[\frac{T-\tau_A}{T}\text{OPT}(\mathcal{P}_{1:T}) + \frac{\tau_A}{T}\text{OPT}(\mathcal{P}_{1:T})\right] \\
&\leq \mathbb{E}\left[(T-\tau_A)\bar{F} + \tau_A\cdot\max_{\boldsymbol{x}\in\mathcal{X}}\left(\widetilde{F}(\boldsymbol{x}) + \boldsymbol{\lambda}^\top\widetilde{\boldsymbol{G}}(\boldsymbol{x})\right)\right] + (\bar{F} + \bar{G}K\frac{\bar{F}}{\beta})\kappa + (\bar{F} + \bar{G}K\frac{\bar{F}}{\beta})\cdot T\delta \\
&= \mathbb{E}\left[(T-\tau_A)\bar{F} + \max_{\boldsymbol{x}\in\mathcal{X}}\sum_{t\in[\tau_A]}\left(\widetilde{F}(\boldsymbol{x}) + \boldsymbol{\lambda}_t^\top\widetilde{\boldsymbol{G}}(\boldsymbol{x})\right)\right] + (\bar{F} + \bar{G}K\frac{\bar{F}}{\beta})\kappa + (\bar{F} + \bar{G}K\frac{\bar{F}}{\beta})\cdot T\delta \\
&\leq \mathbb{E}\left[(T-\tau_A)\bar{F} + \max_{\boldsymbol{x}\in\mathcal{X}}\sum_{t\in[\tau_A]}\left(\hat{F}_{t+\kappa}(\boldsymbol{x}) + \boldsymbol{\lambda}_t^\top\hat{\boldsymbol{G}}_{t+\kappa}(\boldsymbol{x})\right)\right] + (\bar{F} + \bar{G}K\frac{\bar{F}}{\beta})\kappa + 2(\bar{F} + \bar{G}K\frac{\bar{F}}{\beta})\cdot T\delta \\
&= \mathbb{E}\left[(T-\tau_A)\bar{F} + \max_{\boldsymbol{x}\in\mathcal{X}}\mathbb{E}\sum_{t\in[\tau_A]}(\hat{F}_{t+\kappa}(\boldsymbol{x}) + \boldsymbol{\lambda}_{t+\kappa}^\top\hat{\boldsymbol{G}}_{t+\kappa}(\boldsymbol{x}) + (\boldsymbol{\lambda}_t - \boldsymbol{\lambda}_{t+\kappa})^\top\hat{\boldsymbol{G}}_{t+\kappa}(\boldsymbol{x}_t))\right] \\
&\quad + (\bar{F} + \bar{G}K\frac{\bar{F}}{\beta})\kappa + 2(\bar{F} + \bar{G}K\frac{\bar{F}}{\beta})\cdot T\delta \\
&\overset{(a)}{\leq} \mathbb{E}\left[(T-\tau_A)\bar{F} + \max_{\boldsymbol{x}\in\mathcal{X}}\sum_{t\in[\tau_A]}(\hat{F}_{t+\kappa}(\boldsymbol{x}) + \boldsymbol{\lambda}_{t+\kappa}^\top\hat{\boldsymbol{G}}_{t+\kappa}(\boldsymbol{x}_t))\right] + \kappa\eta TK\bar{G}^2 \\
&\quad + (\bar{F} + \bar{G}K\frac{\bar{F}}{\beta})\kappa + 2(\bar{F} + \bar{G}K\frac{\bar{F}}{\beta})\cdot T\delta \\
&\leq \mathbb{E}\left[(T-\tau_A)\bar{F} + \max_{\boldsymbol{x}\in\mathcal{X}}\sum_{t\in[\tau_A-\kappa]}(\hat{F}_{t+\kappa}(\boldsymbol{x}) + \boldsymbol{\lambda}_{t+\kappa}^\top\hat{\boldsymbol{G}}_{t+\kappa}(\boldsymbol{x}_t))\right] + \kappa\eta TK\bar{G}^2 \\
&\quad + 2(\bar{F} + \bar{G}K\frac{\bar{F}}{\beta})\kappa + 2(\bar{F} + K\frac{\bar{F}}{\beta})\cdot T\delta \\
&\overset{(b)}{\leq} \mathbb{E}\left[(T-\tau_A)\bar{F} + \max_{\boldsymbol{x}\in\mathcal{X}}\sum_{t=\kappa+1}^{\tau_A} h_t(\boldsymbol{x})\right] + \kappa\eta TK\bar{G}^2 + 2(\bar{F} + \bar{G}K\frac{\bar{F}}{\beta})\kappa + 2(\bar{F} + \bar{G}K\frac{\bar{F}}{\beta})\cdot T\delta \\
&\leq \mathbb{E}\left[(T-\tau_A)\bar{F} + \max_{\boldsymbol{x}\in\mathcal{X}}\sum_{t\in[\tau_A]} h_t(\boldsymbol{x})\right] + \kappa\eta TK\bar{G}^2 + 2(\bar{F} + \bar{G}K\frac{\bar{F}}{\beta})\kappa + 2(\bar{F} + \bar{G}K\frac{\bar{F}}{\beta})\cdot T\delta
\end{aligned}
$$

$$\overset{(c)}{\leq} \mathbb{E}\Big[(T - \tau_A)\bar{F} + \max_{\boldsymbol{x} \in \mathcal{X}} \sum_{t \in [\tau_A]} h_t(\boldsymbol{x})\Big] + \kappa\eta TK\bar{G}^2 + 2(\bar{F} + \bar{G}K\frac{\bar{F}}{\beta})\kappa + 2R(\bar{F} + \bar{G}K\frac{\bar{F}}{\beta}),$$

where in (a), from (9) in Algorithm 1, we know that $\|\boldsymbol{\lambda}_{t+1} - \boldsymbol{\lambda}_t\|_1 \leq \eta\bar{G}K$, which further implies $\|\boldsymbol{\lambda}_{t+\kappa} - \boldsymbol{\lambda}_t\|_1 \leq \kappa\eta\bar{G}K$ and thus

$$(\boldsymbol{\lambda}_t - \boldsymbol{\lambda}_{t+\kappa})^\top \hat{\boldsymbol{G}}_{t+\kappa}(\boldsymbol{x}_t) \leq \kappa\eta K\bar{G}^2 . \tag{35}$$

In (b), we used the fact that for any $t \geq \kappa + 1$, we have

$$\begin{aligned}
&\mathbb{E}\Big[\max_{\boldsymbol{x} \in \mathcal{X}} \sum_{t \in [\tau_A - \kappa]} (\hat{F}_{t+\kappa}(\boldsymbol{x}) + \boldsymbol{\lambda}_{t+\kappa}^\top \hat{\boldsymbol{G}}_{t+\kappa}(\boldsymbol{x}_t))\Big] \\
&= \mathbb{E}\Big[\max_{\boldsymbol{x} \in \mathcal{X}} \sum_{t \in [\tau_A - \kappa]} \mathbb{E}\Big[h_{t+\kappa}(\boldsymbol{x}) \mid (f_\tau, \boldsymbol{g}_\tau)_{\tau \in [t-1]}\Big]\Big] \\
&\leq \mathbb{E}\Big[\max_{\boldsymbol{x} \in \mathcal{X}} \sum_{t \in [\tau_A - \kappa]} h_{t+\kappa}(\boldsymbol{x})\Big] .
\end{aligned} \tag{36}$$

In (c) we used the fact that $\kappa \geq \log(T)$, so $\delta = R\exp(-\kappa) \geq R$.

Finally, we complete the proof by using the definition $f_t(\boldsymbol{x}_t) = h_t(\boldsymbol{x}_t) - \boldsymbol{\lambda}_t^\top \boldsymbol{g}_t(\boldsymbol{x}_t)$ and following the same argument as in Eq. (22) for the stochastic regime.

### A.8 Proof of Theorem 4.2

*Proof.* We bound the regret in every world as followed

$$\begin{aligned}
\mathcal{R}_T &= \text{OPT}(\mathcal{P}_{1:T}) - \sum_{t \in [T]} \mathbb{E}\left[f_t(\boldsymbol{x}_t)\right] \\
&\overset{(a)}{\leq} \mathbb{E}\Big[\bar{F}(T - \tau_A) + \sum_{t \in [\tau_A]} \boldsymbol{\lambda}_t^\top \boldsymbol{g}_t(\boldsymbol{x}_t) + \mathcal{R}_{\text{BOCO}}(\tau_A)\Big] \\
&\overset{(b)}{\leq} \mathbb{E}\Big[\mathcal{R}_{\text{BOCO}}(\tau_A)\Big]
\end{aligned}$$

where (a) follows from Lemma 4.3, and (b) follows from Lemma 4.5. Recall $\mathcal{R}_{\text{BOCO}}(\tau_A)$ is specified in Lemma 4.3 for each world.

In the following we bound $\mathcal{R}_{\text{BOCO}}(\tau_A)$ for each world.

**Stochastic.**

$$\mathbb{E}\Big[\mathcal{R}_{\text{BOCO}}(\tau_A)\Big] = \mathbb{E}\Big[\max_{\boldsymbol{x} \in \mathcal{X}} \sum_{t \in [\tau_A]} h_t(\boldsymbol{x}) - h_t(\boldsymbol{x}_t)\Big] \overset{(a)}{\leq} \mathcal{O}\Big(\frac{(\rho + \alpha)T}{\beta} + \frac{1}{\gamma_i} + \frac{\gamma_i KT}{\beta^2 \rho^2} + T\epsilon + \frac{1}{\epsilon}\Big) \overset{(b)}{=} \mathcal{O}\big(T^{\frac{3}{4}}\big),$$
$$\tag{37}$$

where (a) follows from Lemma 4.6 by taking the comparator sequence $\boldsymbol{y}_t = \arg\max_{\boldsymbol{x} \in \mathcal{X}} \sum_{t \in [\tau_A]} h_t(\boldsymbol{x})$ for all $t \in [\tau_A]$ such that $P(\boldsymbol{y}_{1:T}) = 1$, as well as any primal ascent expert $i \in [N]$; (b) follows from taking $\eta = \frac{1}{\sqrt{KT}}$, $\rho = K^{\frac{1}{3}}T^{-\frac{1}{4}}$, $\epsilon = T^{-\frac{1}{2}}$, $\beta = \frac{1}{\log(T)}$, and finally choosing $\gamma_i = K^{-\frac{1}{6}}(1 + DT)^{\frac{1}{2}}T^{-\frac{3}{4}}$. Recall all primal ascent expert stepsizes are $\{\gamma_1 \dots \gamma_N\} = \{2^{-i}K^{-\frac{1}{6}}(1 + DT)^{\frac{1}{2}}T^{-\frac{3}{4}} : i = 0 \dots N - 1\}$.

**$\delta$-corrupted, Periodic, and Ergodic.** The proof is nearly identical with that of the stochastic world in Eq. (37) given that we still consider the comparator sequence $\boldsymbol{y}_t = \arg\max_{\boldsymbol{x} \in \mathcal{X}} \sum_{t \in [\tau_A]} h_t(\boldsymbol{x})$ for all $t \in [\tau_A]$ such that $P(\boldsymbol{y}_{1:T}) = 1$. Hence we will omit the proof.

**Adversarial.** Recall the definition $\widetilde{\boldsymbol{y}}_t = \arg\max_{\boldsymbol{x} \in \mathcal{X}} f_t(\boldsymbol{x}_t) + \boldsymbol{\lambda}_t^\top \boldsymbol{g}(\boldsymbol{x}_t)$. Then we have

$$\mathbb{E}\Big[\mathcal{R}_{\mathrm{BOCO}}(\tau_A)\Big] = \Big(1 - \frac{1}{\xi}\Big)\mathrm{OPT}(\mathcal{P}_{1:T}) + \sum_{t \in [\tau_A]} \mathbb{E}\Big[h_t(\widetilde{\boldsymbol{y}}_t) - h_t(\boldsymbol{x}_t)\Big]$$

$$\leq \mathcal{O}\Big(\frac{(\rho + \alpha)T}{\beta} + \frac{1 + P(\widetilde{\boldsymbol{y}}_{1:T})}{\gamma_i} + \frac{\gamma_i KT}{\beta^2 \rho^2} + T\epsilon + \frac{1}{\epsilon}\Big) \tag{38}$$

$$\leq \Big(1 - \frac{1}{\xi}\Big)\mathrm{OPT}(\mathcal{P}_{1:T}) + \tilde{\mathcal{O}}\Big(\sqrt{1 + P(\widetilde{\boldsymbol{y}}_{1:T})} \cdot T^{\frac{3}{4}}\Big),$$

where we chose the primal ascent stepsize $\gamma_i$ s.t.

$$\frac{1}{2}K^{-\frac{1}{6}}(1 + P(\widetilde{\boldsymbol{y}}_{1:T}))^{\frac{1}{2}}T^{-\frac{3}{4}} \leq \gamma_i \leq K^{-\frac{1}{6}}(1 + P(\widetilde{\boldsymbol{y}}_{1:T}))^{\frac{1}{2}}T^{-\frac{3}{4}} \tag{39}$$

We note that such a $\gamma_i$ must exist because $P(\widetilde{\boldsymbol{y}}_{1:T}) \leq DT$ given all $\widetilde{\boldsymbol{y}}_t \in \mathcal{X}$, so that the largest element in the primal ascent stepsize set, namely $K^{-\frac{1}{6}}(1 + DT)^{\frac{1}{2}}T^{-\frac{3}{4}}$ is larger than the upper bound above, namely $K^{-\frac{1}{6}}(1 + P(\widetilde{\boldsymbol{y}}_{1:T}))^{\frac{1}{2}}T^{-\frac{3}{4}}$.

$\square$

## A.9 Proof for Lemma A.1

*Proof.* Recall the definition $h_t(\boldsymbol{x}) = f_t(\boldsymbol{x}) + \boldsymbol{\lambda}_t^\top \boldsymbol{g}_t(\boldsymbol{x})$ in Eq. (5). Then we have

$$\Big|h_t(\boldsymbol{x}) - h_t(\boldsymbol{x}')\Big| \leq \Big|f_t(\boldsymbol{x}) - f_t(\boldsymbol{x}')\Big| + \|\boldsymbol{\lambda}_t\| \cdot \|\boldsymbol{g}_t(\boldsymbol{x}) - \boldsymbol{g}_t(\boldsymbol{x}')\|$$

$$\overset{(a)}{\leq} L\|\boldsymbol{x} - \boldsymbol{x}'\| + K\frac{\bar{F}}{\beta}L\|\boldsymbol{x} - \boldsymbol{x}'\| = (1 + K\frac{\bar{F}}{\beta})L \cdot \|\boldsymbol{x} - \boldsymbol{x}'\|, \tag{40}$$

where (a) follows from the fact that any $(f, \boldsymbol{g}) \in \mathcal{S}$ are $L$-lipschitz under Assumption 2.1.

On the other hand, recall the definition $\hat{h}_t(\boldsymbol{x}) = \mathbb{E}_{\boldsymbol{v} \sim U(\mathbb{B})}[\mathcal{L}_t(\boldsymbol{x} + \rho\boldsymbol{v}, \boldsymbol{\lambda}_t)]$ in Eq. (11). Then we have

$$\Big|h_t(\boldsymbol{x}) - \hat{h}_t(\boldsymbol{x})\Big| = \mathbb{E}_{\boldsymbol{v} \sim U(\mathbb{B})}\Big[h_t(\boldsymbol{x}) - h_t(\boldsymbol{x} + \rho\boldsymbol{v})\Big] \leq (1 + K\frac{\bar{F}}{\beta})L\rho \cdot \mathbb{E}_{\boldsymbol{v} \sim U(\mathbb{B})}\Big[\|\boldsymbol{v}\|\Big] \leq (1 + K\frac{\bar{F}}{\beta})L\rho, \tag{41}$$

where the inequality follows from the first part of this lemma. $\square$

## A.10 Proof of Lemma A.2

*Proof.* Recall the definitions $h_t(\boldsymbol{x}) = f_t(\boldsymbol{x}) + \boldsymbol{\lambda}_t^\top \boldsymbol{g}_t(\boldsymbol{x})$ in Eq. (5), and $\hat{h}_t(\boldsymbol{x}) = \mathbb{E}_{\boldsymbol{v} \sim U(\mathbb{B})}[\mathcal{L}_t(\boldsymbol{x} + \rho\boldsymbol{v}, \boldsymbol{\lambda}_t)]$ in Eq. (11). Then, we have

$$\hat{h}_t(\boldsymbol{y}) - \hat{h}_t(\widetilde{\boldsymbol{x}}_t) \overset{(a)}{\leq} \langle \nabla\hat{h}_t(\widetilde{\boldsymbol{x}}_t), \boldsymbol{y} - \widetilde{\boldsymbol{x}}_t \rangle$$

$$\overset{(b)}{=} \Big\langle \frac{d}{\rho} \cdot \mathbb{E}_{\boldsymbol{u} \sim U(\mathbb{S})}[h_t(\widetilde{\boldsymbol{x}}_t + \rho\boldsymbol{u}) \cdot \boldsymbol{u}], \boldsymbol{y} - \widetilde{\boldsymbol{x}}_t \Big\rangle$$

$$= \mathbb{E}_{\boldsymbol{u}_t \sim U(\mathbb{S})}\Big[\Big\langle \frac{d}{\rho} \cdot h_t(\widetilde{\boldsymbol{x}}_t + \rho\boldsymbol{u}_t) \cdot \boldsymbol{u}_t, \boldsymbol{y} - \widetilde{\boldsymbol{x}}_t \Big\rangle\Big] \tag{42}$$

$$\overset{(c)}{=} \mathbb{E}_{\boldsymbol{u}_t \sim U(\mathbb{S})}[\langle \nabla_t, \boldsymbol{y} - \widetilde{\boldsymbol{x}}_t \rangle]$$

$$\overset{(d)}{=} \mathbb{E}_{\boldsymbol{u}_t \sim U(\mathbb{S})}[\ell_t(\widetilde{\boldsymbol{x}}_t) - \ell_t(\boldsymbol{y})]$$

where (a) follows from concavity of $\hat{h}_t(\cdot)$; (b) follows from Lemma B.2 by taking $h = -h_t$, so that in the lemma $-\nabla_{\boldsymbol{x}}\mathbb{E}_{\boldsymbol{v} \sim U(\mathbb{B})}[h(\boldsymbol{x} + \rho\boldsymbol{v})] = \nabla\hat{h}_t(\boldsymbol{x})$ and $-\mathbb{E}_{\boldsymbol{u} \sim U(\mathbb{S})}[h(\boldsymbol{x} + \rho\boldsymbol{u}) \cdot \boldsymbol{u}] = \mathbb{E}_{\boldsymbol{u} \sim U(\mathbb{S})}[h_t(\boldsymbol{x} + \rho\boldsymbol{u}) \cdot \boldsymbol{u}]$; (c) follows from the gradient estimate in Eq. (7) where

$$\nabla_t = \frac{d}{\rho}\big(f_t(\boldsymbol{x}_t) + \boldsymbol{\lambda}_t^\top \boldsymbol{g}_t(\boldsymbol{x}_t)\big) \cdot \boldsymbol{u}_t = \frac{d}{\rho} \cdot h_t(\boldsymbol{x}_t) \cdot \boldsymbol{u}_t = \frac{d}{\rho} \cdot h_t(\widetilde{\boldsymbol{x}}_t + \rho\boldsymbol{u}_t) \cdot \boldsymbol{u}_t$$

Finally, (d) follows from the definition of surrogate loss functions in Eq. (8). $\square$

## A.11 Proof of Lemma A.3

**Proving (i):**

*Proof.* Let's denote $\nabla_t = \|\boldsymbol{\nabla}_t\|$. Since $\widetilde{\boldsymbol{x}}_{t+1}^i = \Pi_{(1-\alpha)\mathcal{X}}(\widetilde{\boldsymbol{x}}_t^i + \gamma_i \boldsymbol{\nabla}_t)$ we have $\|\boldsymbol{y} - \widetilde{\boldsymbol{x}}_{t+1}^i\| \leq \|\boldsymbol{y} - (\widetilde{\boldsymbol{x}}_t^i + \gamma_i \boldsymbol{\nabla}_t)\|$ for any $\boldsymbol{y} \in (1-\alpha)\mathcal{X}$. Then

$$\|\boldsymbol{y} - \widetilde{\boldsymbol{x}}_{t+1}^i\|^2 \leq \|\boldsymbol{y} - \widetilde{\boldsymbol{x}}_t^i\|^2 - 2\gamma_i \boldsymbol{\nabla}_t^\top (\boldsymbol{y} - \widetilde{\boldsymbol{x}}_t^i) + \gamma_i^2 \nabla_t^2$$
$$\implies \|\widetilde{\boldsymbol{x}}_{t+1}^i\|^2 \leq \|\widetilde{\boldsymbol{x}}_t^i\|^2 + 2\boldsymbol{y}^\top (\widetilde{\boldsymbol{x}}_{t+1}^i - \widetilde{\boldsymbol{x}}_t^i) - 2\gamma_i \boldsymbol{\nabla}_t^\top (\boldsymbol{y} - \widetilde{\boldsymbol{x}}_t^i) + \gamma_i^2 \nabla_t^2 .$$

Hence by taking $\boldsymbol{y} = (1-\alpha)\boldsymbol{y}_t \in (1-\alpha)\mathcal{X}$ and rearranging we get

$$\begin{aligned} & 2\gamma_i \left( \ell_t \left( \widetilde{\boldsymbol{x}}_t^i \right) - \ell_t \left( (1-\alpha)\boldsymbol{y}_t \right) \right) \\ &= 2\gamma_i \boldsymbol{\nabla}_t^\top ((1-\alpha)\boldsymbol{y}_t - \widetilde{\boldsymbol{x}}_t^i) \\ &\leq \|\widetilde{\boldsymbol{x}}_t^i\|^2 - \|\widetilde{\boldsymbol{x}}_{t+1}^i\|^2 + 2(1-\alpha)\boldsymbol{y}_t^\top (\widetilde{\boldsymbol{x}}_{t+1}^i - \widetilde{\boldsymbol{x}}_t^i) + \gamma_i^2 \nabla_t^2 . \end{aligned} \tag{43}$$

Telescoping with $\tau = 1 \ldots t$ we get

$$\begin{aligned} & \sum_{\tau \in [t]} \ell_\tau \left( \widetilde{\boldsymbol{x}}_\tau^i \right) - \sum_{\tau \in [t]} \ell_\tau \left( (1-\alpha)\boldsymbol{y}_\tau \right) \\ &\leq \frac{1}{2\gamma_i} \|\widetilde{\boldsymbol{x}}_1^i\|^2 + \frac{1-\alpha}{\gamma_i} \sum_{\tau \in [t]} \boldsymbol{y}_\tau^\top (\widetilde{\boldsymbol{x}}_{\tau+1}^i - \widetilde{\boldsymbol{x}}_\tau^i) + \frac{\gamma_i}{2} \sum_{\tau \in [t]} \nabla_\tau^2 \\ &= \frac{1}{2\gamma_i} \|\widetilde{\boldsymbol{x}}_1^i\|^2 + \frac{1-\alpha}{\gamma_i} \left( \boldsymbol{y}_t^\top \widetilde{\boldsymbol{x}}_{t+1}^i + \sum_{\tau \in [t-1]} (\boldsymbol{y}_\tau - \boldsymbol{y}_{\tau+1})^\top \widetilde{\boldsymbol{x}}_{\tau+1}^i \right) + \frac{\gamma_i}{2} \sum_{\tau \in [t]} \nabla_\tau^2 \\ &\leq \frac{1}{2\gamma_i} \|\widetilde{\boldsymbol{x}}_1^i\|^2 + \frac{1-\alpha}{\gamma_i} \left( \|\boldsymbol{y}_t\| \cdot \|\widetilde{\boldsymbol{x}}_{t+1}^i\| + \sum_{\tau \in [t-1]} \|\boldsymbol{y}_\tau - \boldsymbol{y}_{\tau+1}\| \cdot \|\widetilde{\boldsymbol{x}}_{\tau+1}^i\| \right) + \frac{\gamma_i}{2} \sum_{\tau \in [t]} \nabla_\tau^2 \\ &\leq \frac{(1-\alpha)^2 D^2}{2\gamma_i} + \frac{(1-\alpha)^2 D}{\gamma_i} \left( P(\boldsymbol{y}_{1:T}) + D \right) + \frac{\gamma_i d^2}{2\rho^2} \left( \bar{F} + K\frac{\bar{F}}{\beta}\bar{G} \right)^2 t , \end{aligned} \tag{44}$$

where $P(\boldsymbol{y}_{1:T})$ is defined in the lemma statement. $\qquad\square$

**Proving (ii):**

*Proof.* First, we have for any $t \in [T], i \in [N]$

$$\left| \ell_t(\widetilde{\boldsymbol{x}}_t^i) \right| = \left| \boldsymbol{\nabla}_t^T (\widetilde{\boldsymbol{x}}_t - \widetilde{\boldsymbol{x}}_t^i) \right| \leq \|\boldsymbol{\nabla}_t\| \cdot \|\widetilde{\boldsymbol{x}}_t^i - \widetilde{\boldsymbol{x}}_t\| \leq \frac{d}{\rho} \left( \bar{F} + K\frac{\bar{F}}{\beta}\bar{G} \right) \cdot (1-\alpha)D , \tag{45}$$

where we recall $D = \sup_{\{\boldsymbol{x}, \boldsymbol{x}'\} \subseteq \mathcal{X}} \|\boldsymbol{x} - \boldsymbol{x}'\|$ is the diameter of $\mathcal{X}$, and both $\{\widetilde{\boldsymbol{x}}_t^i, \widetilde{\boldsymbol{x}}_t\} \subseteq (1-\alpha)\mathcal{X}$.

Define $W_t = \sum_{i \in [N]} w_{i,t}$ for all $t \in [T]$, then

$$\begin{aligned} \log\left( \frac{W_{t+1}}{W_t} \right) &= \log\left( \sum_{i \in [N]} \frac{w_{i,t} \exp\left( -\epsilon \ell_t(\widetilde{\boldsymbol{x}}_t^i) \right)}{W_t} \right) \\ &= \log\left( \mathbb{E}_{I_t \sim \boldsymbol{w}_t/W_t} \left[ \exp\left( -\epsilon \ell_t(\widetilde{\boldsymbol{x}}_t^{I_t}) \right) \right] \right) \\ &\overset{(a)}{\leq} -\epsilon \mathbb{E}_{I_t \sim \boldsymbol{w}_t/W_t} \left[ \ell_t(\widetilde{\boldsymbol{x}}_t^{I_t}) \right] + \frac{\epsilon^2}{2} \cdot \left( \frac{d}{\rho} \left( \bar{F} + K\frac{\bar{F}}{\beta}\bar{G} \right) \cdot (1-\alpha)D \right)^2 \\ &\overset{(b)}{=} -\epsilon \ell_t \left( \mathbb{E}_{I_t \sim \boldsymbol{w}_t/W_t} \left[ \widetilde{\boldsymbol{x}}_t^{I_t} \right] \right) + \frac{\epsilon^2}{2} \cdot \left( \frac{d}{\rho} \left( \bar{F} + K\frac{\bar{F}}{\beta}\bar{G} \right) \cdot (1-\alpha)D \right)^2 \\ &\overset{(c)}{=} -\epsilon \ell_t \left( \widetilde{\boldsymbol{x}}_t \right) + \frac{\epsilon^2}{2} \cdot \left( \frac{d}{\rho} \left( \bar{F} + K\frac{\bar{F}}{\beta}\bar{G} \right) \cdot (1-\alpha)D \right)^2 . \end{aligned} \tag{46}$$

Here (a) follows from Hoeffding's Lemma as described in Lemma B.1 where we take $X = \ell_t(\widetilde{\boldsymbol{x}}_t^{I_t})$, $a = -\frac{d}{\rho}\left(\bar{F} + K\frac{\bar{F}}{\beta}\bar{G}\right) \cdot (1-\alpha)D$ and $b = \frac{d}{\rho}\left(\bar{F} + K\frac{\bar{F}}{\beta}\bar{G}\right) \cdot (1-\alpha)D$; (b) follows from the definition that $\ell_t(\widetilde{\boldsymbol{x}}) = \boldsymbol{\nabla}_t^T(\widetilde{\boldsymbol{x}} - \widetilde{\boldsymbol{x}}_t)$ is a linear function in $\widetilde{\boldsymbol{x}}$; (c) follows from Eq.(6).

Hence, telescoping the above we get

$$\log\left(\frac{W_{t+1}}{W_1}\right) \leq -\epsilon \sum_{\tau \in [t]} \ell_\tau(\widetilde{\boldsymbol{x}}_\tau) + \frac{t\epsilon^2}{2} \cdot \left(\frac{d}{\rho}\left(\bar{F} + K\frac{\bar{F}}{\beta}\bar{G}\right) \cdot (1-\alpha)D\right)^2. \qquad (47)$$

On the other hand, we have

$$
\begin{aligned}
\log\left(\frac{W_{t+1}}{W_1}\right) &= \log(W_{t+1}) - \log(W_1) \\
&\geq \log(\max_{i \in [N]} w_{i,t}) - \log(N) \\
&= \max_{i \in [N]} \log(w_{i,t}) - \log(N) \\
&\overset{(a)}{=} \max_{i \in [N]} \log\left(w_{i,1} \exp\left(-\epsilon \sum_{\tau \in [t]} \ell_\tau(\widetilde{\boldsymbol{x}}_\tau^i)\right)\right) - \log(N) \\
&= -\epsilon \min_{i \in [N]} \sum_{\tau \in [t]} \ell_\tau(\widetilde{\boldsymbol{x}}_\tau^i) - \log(N).
\end{aligned}
\qquad (48)
$$

Hence, after combining Eqs. (47) and (48), and dividing both sides by $\epsilon > 0$, we get

$$-\sum_{\tau \in [t]} \ell_\tau(\widetilde{\boldsymbol{x}}_\tau) + \frac{t\epsilon}{2} \cdot \left(\frac{d}{\rho}\left(\bar{F} + K\frac{\bar{F}}{\beta}\bar{G}\right) \cdot (1-\alpha)D\right)^2 \geq -\min_{i \in [N]} \sum_{\tau \in [t]} \ell_\tau(\widetilde{\boldsymbol{x}}_\tau^i) - \frac{\log(N)}{\epsilon}$$

$$\implies \sum_{\tau \in [t]} \ell_\tau(\widetilde{\boldsymbol{x}}_\tau) - \min_{i \in [N]} \sum_{\tau \in [t]} \ell_\tau(\widetilde{\boldsymbol{x}}_\tau^i) \leq \frac{t\epsilon}{2} \cdot \left(\frac{d}{\rho}\left(\bar{F} + K\frac{\bar{F}}{\beta}\bar{G}\right) \cdot (1-\alpha)D\right)^2 + \frac{\log(N)}{\epsilon}$$

$$\implies \sum_{\tau \in [t]} \ell_\tau(\widetilde{\boldsymbol{x}}_\tau) - \sum_{\tau \in [t]} \ell_\tau(\widetilde{\boldsymbol{x}}_\tau^i) \leq \frac{t\epsilon}{2} \cdot \left(\frac{d}{\rho}\left(\bar{F} + K\frac{\bar{F}}{\beta}\bar{G}\right) \cdot (1-\alpha)D\right)^2 + \frac{\log(N)}{\epsilon}, \quad \forall i \in [N],$$

$$(49)$$

which completes the proof. $\qquad\square$

## B  Supplementary lemmas

**Lemma B.1** (Hoeffding's lemma). *Let $X$ be some random variable such that $a \leq X \leq b$ almost surely for some $a, b \in \mathbb{R}$. Then for any $\epsilon \in \mathbb{R}$, we have $\mathbb{E}\left[\exp(-\epsilon X)\right] \leq \exp\left(-\epsilon \mathbb{E}\left[X\right] + \frac{\epsilon^2(b-a)^2}{8}\right)$.*

**Lemma B.2** ([25] Lemma 2.1). *Let $h : \mathcal{X} \to \mathbb{R}$ be some convex function (not necessarily differentiable). Then for any $\boldsymbol{x} \in \mathcal{X} \subseteq \mathbb{R}^d$ and $\delta > 0$ we have*

$$\nabla_{\boldsymbol{x}}\mathbb{E}_{\boldsymbol{v} \sim U(\mathbb{B})}[h(\boldsymbol{x} + \delta\boldsymbol{v})] = \frac{d}{\delta} \cdot \mathbb{E}_{\boldsymbol{u} \sim U(\mathbb{S})}\left[h(\boldsymbol{x} + \delta\boldsymbol{u}) \cdot \boldsymbol{u}\right]. \qquad (50)$$

