# OpenReview forum: "Online Ad Procurement in Non-stationary Autobidding Worlds"
_NeurIPS.cc/2023/Conference — NeurIPS 2023 poster_

### Official Review · Reviewer_bowu · 2023-07-04

**Soundness:** 2 fair
**Presentation:** 3 good
**Contribution:** 3 good
**Rating:** 7
**Confidence:** 4

**Summary:**

This work studies an advertiser's online high-dimensional lever decision problem with long-tern constraints under limited bandit feedback for different input models. The authors' main contributions include: (1) model formulation; (2) proposing an algorithm universally applicable to input models; (3) theoretical regret analysis.

There is an advertiser that repeatedly interact with an ad platform during a time horizon $T$, aiming to maximize her total conversions subject to multiple constraints. At each time $t$, the advertiser need to make a multi-dimensional lever decision and observed her realized conversion as well as her multi-dimensional realized cost. The authors propose an algorithm with universally good performance and provide regret lower bounds of this problem with respect to different input procedures.

**Strengths:**

1. The proposed algorithm is oblivious to input models such that it can achieve high performance without knowing which setting the decision-maker is in.

2. It is novel to adopt the random perturbation approach and the expert-based decision-making in online autobidding. The regret analysis requires non-trivial insights and techniques. Particularly, the proof of lemma 4.6 that bounds the primal ascent regret stands out from standard primal-dual frameworks.

3. The organization of this paper is great.

**Weaknesses:**

1. The idea of solving a problem in many worlds is not new. In addition to commonly studied stochastic and adversarial input models, [1] also considers $\delta$-corrupted, periodic, ergodic input models. Actually, the proofs for the latter three cases are analogous to the stochastic case as they still assume stationary distributions to some extent.

[1] Santiago Balseiro, Haihao Lu, and Vahab Mirrokni. The best of many worlds: Dual mirror descent for online allocation problems.

2. Some mistakes are spotted. Some statements are not clear enough. See the questions below.

**Questions:**

1. Line 13, multi-dimension -> multi-dimensional.

2. Line 14. What does "uncertain" mean throughout this paper? The bidder should known their constraints at the beginning, e.g. known budget.

3. Line 168, a redundant "of".

4. Line 219, realzied -> realized.

5. Line 283, violates constraints violation?

6. Line 664, missing values for $a$ and $b$.

7. Line 288, what does $\alpha$ do in algorithm 1? To my understanding, it makes sure that the opinion of an expert lies in the interior of the domain. However, the value of $\alpha$ is not specified. It seems fatal if $\alpha$ is close to 1.

8. Line 299, the number of experts $N$ seems to be negative when $T$ goes to infinity.

9. Line 301, N experts or (N+1) experts?

10. Line 539, the first term in the RHS of equation (15) is not bounded after timing $T$.

11. Line 542, $\tau$ -> $t$. The authors should check $t\in [T]$ and $\tau\in [t]$ throughout the appendix. the order of $\beta$ also seems wrong in equation (16).

12. Line 626, observe that the sum operator is over $t\in [\tau_A]$ so the optimal $x$ does not vary to $t$. Does it implies all $y_t$ are the same?

13. Line 629, arer -> are.

14. Line 636, according to the theorem 4.2 one has $\gamma_{0} = K^{-1/6} (1+DT)^{1/2} T^{-3/4}$ and $\gamma_{N} = 1$. Then $\gamma_{0}$ is not the largest element in the stepsize set since it goes to $0$.

Be willing to raise my rating if it is confirmed there is no technical flaw.

---

> ### Author Rebuttal · Authors · 2023-08-10
>
> We thank the reviewer for the constructive and valuable feedback.
>
> Response to weakness: The key difference between our paper and [1] is that the reward function and the constraint functions are known in [1] before making the decision (i.e. [1] studies the full information setting), while in this paper we study the bandit setting where the reward and constraint functions are unknown before a decision is made. This makes the analysis much more difficult than that of [1], as we need to carefully handle unknown per-round constraint violation (before making decisions) to satisfy long-term constraints, while not allowing regret to decay too much. Our proposed algorithm requires much more complex design and analyses than that of [1], which simply employs a mirror descent approach. We would also like to point out that although our proposed algorithm is inspired by well-established approaches, it is interpretable and implementable and thus can be easily adopted in practical setups. We also believe that our proposed algorithm can inspire future works to develop more complex algorithms and improve performance guarantees (to match with lower bounds for the full-information setting).
>
> Response to questions: We apologize for the typos in the paper, and we will carefully review the paper to ensure accuracy of our statements and results. In the following we will address major technical questions raised by the reviewer, namely questions 2, 6, 7, 8, 9, 10, 11, 12, 14:
>
> (2). “Uncertain” refers to the fact that the amount of per-round constraint violation each period is unknown. Take for example a single long-term budget constraint which says total spend cannot exceed a certain amount. In our bandit setting, the incurred cost for making a decision is only observed after the decision is made, and hence there are two sources of uncertainty: (a) The decision maker does not know how much budget is consumed before making any decision; (b) The decision maker may not necessarily satisfy her long-term budget constraint if she does not act intelligently.
>
> (6). In line 664, we utilize Eq. (45) and choose $a = -\frac{d}{\rho}\left(\bar{F}+ K \frac{\bar{F}}{\beta} \bar{G}\right)\cdot (1-\alpha) D$ and $b = \frac{d}{\rho}\left(\bar{F}+ K  \frac{\bar{F}}{\beta}  \bar{G}\right)\cdot (1-\alpha) D$, while the original term of $\frac{\epsilon^2}{8}$ will be replaced by $\frac{\epsilon^2(b-a)^2}{8}$. We remark that the rest of the proof follows with this small correction.
>
> (7). You are absolutely correct that $\alpha$ ensures the opinion of an expert lies in the interior of the domain.  In the input line of Algorithm 1 on page 7, we do require choosing $\alpha \in (0,1)$.  In particular, in the analysis (specifically for Lemma 4.6), we set $\alpha$ to be a parameter with the same order of $\rho$; i.e., $\alpha = K^{1/3}T^{-1/4}$.  We will add the requirements on $\alpha$ explicitly in the statement of Theorem 4.2.
>
> (8). Thank you for catching this typo!  Instead of the previous definition of $N$, we should set $N = \max (1, \lceil  -\log_2 (K^{-\frac{1}{6}}(1+D T)^{\frac{1}{2}}T^{-\frac{3}{4}}) \rceil ) = O(\log(T))$, which is always positive.
>
> (9). Thank you for pointing this out.  We should have $N+1$ experts.
>
> (10).  We take $\alpha  = K^{1/3}T^{-1/4}$ as mentioned in the answer to Question 7. This ensures the regret due to this term after $T$ rounds is $O(T^{3/4})$, which matches our final regret bound.  We will emphasize this explicitly in the revised version of our paper.
>
> (11). Thank you for carefully reading our paper and catching this typo.  We will fix all these typos as the reviewer mentioned, including $(\tau, t)$ and the order of $\beta$ in Equation (16) (which should be squared). Note that this does not impact the final regret bound as $\beta$ is in the order of $1/\log(T)$.
>
> (12). Thank you for your comment.  Yes, all $y_t$ are the same in the stochastic setting, given the nature of i.i.d. randomness. We remark that to the best of our knowledge, it is not clear for the other settings what structural properties the $y_t$ sequences possess, as they may vary significantly depending on the specific (unknown) underlying reward distribution sequences. Mathematically this is one aspect that makes the problem we are tackling extremely challenging. Please see our response to Question 2 from reviewer EvVQ for more detailed discussions.
>
> (14). We believe this issue will be fixed after changing the value of $N$ as we suggested in the answer of Question 8. The primal ascent step sizes will then be decreasing as $i$ increases.
>
> We hope the above response addresses the reviewer’s concerns about technical details, and we would be more than happy to answer other questions should they arise. Given our above responses to technical concerns, we would greatly appreciate it if the reviewer can re-assess the contributions of this submission as well as the corresponding rating. Again, we sincerely thank the reviewer for the comments, suggestions and questions.

---

> > ### Comment · Reviewer_bowu · 2023-08-18
> >
> > Line 634-635. With the new definition of $N_t$, how do you get the result of equation (38) by using equation (39)? By the choice of $\gamma_i$, $$O\left(\frac{1+P}{\gamma_i}\right) = O((1+P)^{1/2} \cdot T^{3/4}) \leq O(T^{1/2} \cdot T^{3/4}),$$ which is greater than $O(T)$.

---

> > > ### Author Response · Authors · 2023-08-20
> > >
> > > We apologize for the lack of clarity here which is related to the earlier typos pointed in the review, and we remark that the final regret bound should indeed be $(1-1/\chi)OPT + T^{3/4}\sqrt{P}$ as pointed out by the reviewer. Despite the fact that when $P$ is at least the order of $\sqrt{T}$ we get $T^{3/4}\sqrt{P} \geq O(T)$ which leads to a non-meaningful regret bound, we remark that to the best of our knowledge even in the no-constraint setting, there does not exist any algorithm that is able to remove the term $T^{3/4}\sqrt{P}$ in the bandit, multi-dimensional, single-point feedback online optimization setup with dynamic regret (see detailed discussion in Remark 4 of [R1]). The most widely studied approach to achieve a sharper bound involves querying twice each round (i.e. two-pointed feedback) but this is not the practical setup we consider in the paper in the context of online advertising. Further, as described in [R1], the term $P$ can be viewed as a problem instance-dependent factor that measures the hardness of the problem. Note that the term $(1-1/\chi)OPT $ is the regret lower bound and unavoidable [R2], and our bound still bears value in the case where $P = o(\sqrt{T})$. Nevertheless, we acknowledge that this adversarial bound is not satisfactory, and we believe that a key future direction would be to improve this bound.
> > >
> > > Finally, we will definitely add more discussions in the paper on the limitations of our bounds to the adversarial setup, and complement the paper with numerical studies to illustrate that our algorithm performs well in practical settings.
> > >
> > > [R1] Zhao, Peng, et al. "Bandit convex optimization in non-stationary environments." The Journal of Machine Learning Research.
> > >
> > > [R2] Balseiro, Santiago R., Haihao Lu, and Vahab Mirrokni. "The best of many worlds: Dual mirror descent for online allocation problems." Operations Research

---

> > > > ### Comment · Reviewer_bowu · 2023-08-21
> > > >
> > > > Thanks for the authors' detailed and careful responses. All my concerns have been clearly addressed.
> > > >
> > > > By the result of [R2], $O(T^{3/4}(1+P)^{1/2})$ is almost the best we can get for one-point feedback. Even under two-point feedback, as the lower bound is $\Omega(\sqrt{TP})$, one has to consider $P=o(T)$ to ensure the convergence of long-term performance in the adversarial setting. By the result of [R2], $\xi$ is the best competitive ratio we can get. Now I regard $(1-1/\xi)\mathrm{OPT} + O(T^{3/4}(1+P)^{1/2})$ as a standard and reasonable regret bound. However, the current submission gives me a feeling that the authors are hiding it on purpose. This technical detail is actually very important and must be written explicitly wherever the bound is mentioned, namely Table 1 and Theorem 4.2, to properly reflect the contribution of this paper. More discussions in section 5 that involve the bounds in [R1]  can help justify this result.
> > > >
> > > > I have decided to give this work a higher rating. But the authors really need to proofread this paper again. I'm sure there must be many other typos that haven't been caught by me or the other reviewers.

---

### Official Review · Reviewer_EvVQ · 2023-07-06

**Soundness:** 3 good
**Presentation:** 2 fair
**Contribution:** 3 good
**Rating:** 6
**Confidence:** 3

**Summary:**

This work studies the problem of dynamic online allocation under constraints with bandit feedback, and derives a generic algorithm applicable to various input settings (stochastic, adversarial, $\delta$-corrupted, ergodic, periodic). It recovers regret rates close to those of the lower bounds in each of these settings. The algorithm uses a dual gradient descent over $\lambda_t$ to decouple the decisions over time by considering the lagrangian, a gradient ascent for the optimal choice of $x_t$ (that also uses the technique from Flaxman et al (2004) to handle the bandit feedback), and finally a multiplicative weight update to adapt the learning rates used to the correct input setting.

**Strengths:**

- The algorithm generalizes to multiple input settings and general constraints the problems related to Bandits with knapsack constraints, and of online learning with constraints. (to be clear this is an important strength of this paper!)
- The problem is well motivated through the consideration of running multiple ad campaigns


**Weaknesses:**

- The assumption that there exists some safe action of level $\beta$ do simplify the problem of constraint satisfaction by guaranteeing to be able to satisfy the constraint at the end
- Some of the upper bounds (stochastic, periodic, corrupted) are a bit loose compared to their respective lower bounds
- The writing can be improved, in particular in the proofs which are lacking discussion about their main ideas and intuition. As an example page $8$ of the appendices is almost only a sequence of inequality and is hard to follow. For instance the inequalities could be cut in multiple parts, and some comments could be added to better explain the goal of the proof. There are also some typos.


**Questions:**

- Do the $T^{2/3}$ rates come from the use of the method to handle bandit feedback? How would a generic gradient feedback affect the rates, would we be able to derive tight rates with respect to the lower bounds presented? I think it might have been better to first present the results with gradient feedback, and mention in the appendix that bandit feedback can be handled by a standard technique. This would allow the main part of the paper to focus on the new contributions.
- Could you give some intuition on what this optimal dynamic sequence looks like in the various settings? For instance, if my understanding is correct, the optimal dynamic sequence in the stochastic setting is simply a unique point (because the data is i.i.d). Could the optimal sequence for the ergodic setting be a function of $\kappa$ close to the unique optimal point with respect to the stationary measure?
- How much does the meta algorithm degrade the regret? (compared to assuming that we know the input setting)
- l 657 Why is it $... +D \dots$ and not $...+(t-1)D\dots $ ?

Comments/Typos:
- L304 and l324 stochstic ->  stochastic
- I find the notation of $\lambda \in [0,F e/ \beta]$ confusing, is it $\lambda \in [0,F/\beta]^K$?
- L553 ‘which states $\max_{x \in \mathcal{X}} f(x)$: is a word missing?
- L608 I am not sure to understand this statement, could you include a reference for this result? Is the sup taken over $(g_{\tau},f_{\tau})$ - over $[t]$ or $[T]$?
- I think it would be nice to cite the paper of Mannor et al (Online learning with sample path constraints), which deals with very similar problem and started the works on online convex optimization with varying constraints
- Equation below l643: $v$ -> $\Vert v \Vert$, and the last equality should be an inequality (as it is in the ball, not the sphere)
- Equation below l655 I think some of the gradients are missing $\Vert \nabla_t \Vert^2$ and $x^i_{t+}$ ->  $\tilde{x}^{i}_{t}$. Why are some of the gradients bolded and not the others?
- Same thing for the gradients below l657, in addition I am not sure where $P(y_1:T)$ is defined.
- L 664 ‘$a=$ and $b=$’ it is unfinished

I have not read through all the proofs, but I would recommend to read it again to look for additional typos


**Limitations:**

Yes the authors address some limitations in the section $5$ of the paper regarding the lower bounds.

---

> ### Author Rebuttal · Authors · 2023-08-10
>
> We thank the reviewer for the constructive and valuable feedback.
>
> Regarding Weakness 1 on the safe action: We would like to point out that the existence of a “safe action” is quite common in online advertising. Take for example the simple case where an advertiser only has a long-term budget constraint, and the only lever used is the budget set each round (i.e. a single decision variable that represents the maximum spend in each round/ad campaign). In this case, the safe action is to simply set the per-round budget to be 0, so the advertiser would always acquire some positive constraint balance (i.e. limit spend to ensure expenditure does not exceed long-term budget after $T$ rounds). Another example is the simple case where the advertiser only has a long-term ROI constraint which states total reward exceeds total spend after $T$ rounds, and again assume the single decision variable of interest is per-round budget. Then, by setting a small per-round budget, ad platforms tend to procure the most “lucrative ads” on behalf of the advertiser that have a large value-to-cost ratio, which generates a positive ROI-balance per round that helps satisfy the long-term ROI constraint after $T$ rounds. Similar assumptions are made in many related works see; e.g. [R1,R2]. Without the safe action, there is no guarantee that the problem is feasible and performance guarantees may be intractable.
>
> [R1] Deng, Yuan, et al. "Multi-channel autobidding with budget and roi constraints."
>
> [R2] Feng, Zhe, Swati Padmanabhan, and Di Wang. "Online Bidding Algorithms for Return-on-Spend Constrained Advertisers."
>
> Regarding Weakness 2 on loose upper bounds: We would like to point out that our setting is related to the more general problem of bandit online constrained optimization in continuous high-dimensions with single-point feedback (i.e. the decision maker can only make a decision once per round and observe a single feedback). To the best of our knowledge, there is no existing algorithm that achieves optimal bounds even in purely stochastic or adversarial environments (see discussions in [R3]). Our paper considers an even more complex setup than these “pure environments” as we demand a single algorithm that yields tractable performance across various non-stationary environments. Indeed, many of the lower bound stated in the paper are for a more restrictive class of problems under full-information feedback (i.e., the reward and constraint functions are known before making any decision, while we study bandit feedback). To the best of our knowledge, lower bounds for bandit feedback problems are unknown, so we present those for full-information feedback. The gap may come from the intrinsic difference between these two classes of problems. Nevertheless, we believe that this work has the potential to inspire future research to develop more sophisticated algorithms and close the upper-lower bound gap.
>
> [R3] Zhao, Peng, et al. "Bandit convex optimization in non-stationary environments."
>
> Regarding Weakness 3: we thank the reviewer for pointing this out and agree that the exposition of the paper can be improved. We will revise our paper and include more discussions accordingly in both the main body and the technical results/proofs in the appendix.
>
> Response to Question 1: The presented non-optimal rates are indeed due to the need to handle bandit feedback. In the full-information setup where the decision maker can first observe the reward and constraints in each period before making a decision,  it has been shown in [R4] that a simple mirror descent approach can achieve optimal rates. Nevertheless, we point out that the bandit setup considered in this paper is much more complex and requires our proposed techniques to effectively balance regret minimization and constraint satisfaction in a limited information bandit environment. See the response for weakness 2 for more details.
>
> [R4] Balseiro, Santiago R., Haihao Lu, and Vahab Mirrokni. "The best of many worlds: Dual mirror descent for online allocation problems."
>
> Response to Question 2: The reviewer’s insight is correct that in the stochastic setting the optimal dual variable sequence is a fixed point, and the optimal primal variables of course would depend on the observation. Nevertheless, to the best of our knowledge, it is not clear for the other settings what structural properties the optimal sequences possess, and the optimal sequence may vary significantly depending on the specific (unknown) underlying reward distribution sequences. Take an extremely simplistic example in the $\delta$-corrupted environment, where there is only a single round at which the reward distribution is corrupted and the advertiser only has a long-term budget constraint. If the corrupted expected reward is very large (e.g. in the order of $T$), then she should spend most of her budget during that single round in the optimal decision sequence, and spend nearly nothing in other rounds; on the other hand, if the reward in the corrupted round is 0, then the setting reduces to the stochastic setting, and the optimal action in each round should be some fixed point. Nevertheless, we believe we can produce experimental simulations to characterize the structure of optimal sequences in simple settings, and we would include relevant discussions/results in the revised version of our paper.
>
> Response to Question 3:
> We will include a summary of the best existing upper bounds for each non-stationary environment, respectively, in the paper in our revision.
>
> Response to Question 4: we apologize for the typo in line 657. In the second equality (within the large parentheses), we should have $y_{t}^\top\tilde{x}_{t+1}^i$ (which is independent of $\tau$ that we sum over). This term can then be bounded by $D$ (as opposed to $(t-1)D$).
>
> Finally, we thank the reviewer for the comments and for pointing out typos. We will carefully review the paper and make corrections accordingly.

---

> ### Comment · Reviewer_EvVQ · 2023-08-13
>
> I thank the authors for their detailed and careful responses.
>
> If I understand correctly the reply regarding the safe action, this assumes that the budget scales with $T$ (e.g. $B=\rho T$), so that $g_t(x_t)=\text{cost}_t*x_t-\rho$ and thus $g_t(0)=-\rho<0$? If so then indeed it is reasonable; I think it would be nice to include the examples mentioned by the authors to justify the existence of such a safe action.
>
> I think the novelty of the technical contributions would be clearer if in the proof sketches this work was more directly compared to [9] and its proof techniques, as well as which arguments need special care in the combination of the bandit gradient estimation technique and the mirror descent for $\lambda_t$.
>
> Many typos were found in the proofs (in particular by reviewer bowu), and as such I will keep my current grade.
>
> Otherwise, all my questions have been clearly addressed.
>
> I believe that the paper should be accepted, assuming that the comparison with [9] is made clearer, and that this work is proofread again (including the appendix) to catch any additional typos.

---

> > ### Author Response · Authors · 2023-08-15
> >
> > Regarding your understanding/example for the safe action (in the case of budgets), you are completely correct. We will indeed include such examples in the revision of the paper to provide intuition for existence of the safe action.  We will also present more comparisons with [9] in our revision to clearly convey our contributions and novelties, and will carefully review the paper to correct typos.
> >
> > Again, we sincerely thank the reviewer for taking the time to offer all the constructive feedback!

---

### Official Review · Reviewer_hupT · 2023-07-06

**Soundness:** 4 excellent
**Presentation:** 4 excellent
**Contribution:** 3 good
**Rating:** 7
**Confidence:** 4

**Summary:**

This paper concerns a two-stage autobidding scenario, such as an advertising platform environment.  Each advertiser wants to maximize value received (e.g., clicks) subject to long-run constraints (e.g., budget or ROI).  As actions, the advertiser can specify certain instructions to an autobidding agent (e.g., a spend rate or ROAS target) and observe the results.  The advertiser observes bandit feedback, and wishes to tune their choice of actions/instructions to solve their long-run optimization problem.

Because the autobidders themselves are learning over time and potentially facing changing market conditions, the evolution of payoffs observed by the advertiser may not be stationary.  The paper considers a variety of different payoff evolution models, including partial adversarial corruption, periodic, and ergodic payoffs.

The main result is a universal learning method for the advertiser that achieves good regret for each of these payoff evolution models.  The idea is to combine dual descent methods for constraints with a modified online convex optimization approach to adequately explore lever decisions given the dual variables.  This combination leads to vanishing regret in each of the settings considered.  The resulting regret rates are then compared with known lower bounds for each of these settings (many of which apply in relaxed settings, such a full feedback rather than bandit feedback).

**Strengths:**

I like this paper.  The modeling framework that separates true "long-run" objectives from "short-term" directive levers is extremely natural and, as far as I'm aware, novel.  It also tracks my understanding of how autobidding works in practice: advertisers need not keep their specified constraints (to the autobidder) fixed over time, but can manipulate them online as tunable knobs.

The assumption of a safe action is likewise very reasonable, and the proposed algorithm makes use of it to good effect.

The proposed algorithm combines multiple well-established ideas from the online optimization literature in a reasonable way.  The fact that this comes together into a unifying framework is an appealing feature, as is the need for only bandit feedback.  The regret rate suffers somewhat compared to known bounds, but not by too much --- what amounts to a rate of T^{3/4} for each of the non-fully-adversarial settings is quite good, while leaving room for future work to improve.  Getting these regret rates down to sqrt{T} (or showing this isn't possible) is a nice open challenge.


**Weaknesses:**

The biggest question for me is how "real" autobidding (e.g., competitive uniform bidding) falls into this scenario.  For instance: even in a stationary environment in terms of competitors, the relationship between levers and outcomes is not necessarily stationary for advertiser i because the underlying autobidder is learning over time how best to satisfy the directive communicated by a given lever setting.  So my understanding is that this scenario would fall under the ergodic setting.  But what if the competitors are not stationary, but are learning as well?  Can these theoretical frameworks be linked back to the motivating setting of autobidders that simultaneously learn?  Either way, it would be nice to have a more thorough discussion of this in the body of the paper.

Another potential weakness is that the technical contributions largely synthesizes known approaches, so the marginal technical contribution is not extremely high.  I therefore view the conceptual and modeling contributions as the main selling points for the submission.

**Questions:**

Are there natural conditions under which a scenario with mutually competing autobidders would fall into one of the analyzed worlds?

**Limitations:**

I feel that the paper is sufficiently up-front about its limitations, and the authors do an adequate job of describing what their paper does and does not do.

---

> ### Author Rebuttal · Authors · 2023-08-10
>
> We thank the reviewer for the constructive and positive feedback!
>
> Regarding Weakness 1 and Question 1: The game-theoretic interactions between actions among multiple agents is indeed an interesting yet challenging future direction. For this work, one can view the various environments of interest (namely stochastic, adversarial, periodic, Markovian, and corrupted) as “aggregate market-dynamics” driven by competitor algorithms. We believe that this is a reasonable aggregate view of real-world algorithmic bidding in online advertising markets: for example, the environment is mostly stationary (or i.i.d) when looking at a short time period, such as one hour (see [R1] for practical evidence); for longer time horizons, aggregate algorithmic interactions may exhibit periodicity (such as more aggressive and active bidding during non-work hours) see [R2] for practical evidence; competing algorithms may occasionally be driven by adversarial behavior due to the competition [R3] etc.. The beauty of the proposed algorithm is that the algorithm can achieve good performance without the need of knowing this “aggregate view" on competitor algorithmic behaviors. We would also like to point out that most of the previous theoretical works on online learning in advertising focus on either stochastic i.i.d. model, which is too optimistic in practice, or adversarial model, which is too pessimistic. Nevertheless, we completely agree that our paper can benefit from including discussions on relevant multi-agent learning topics, and we will also support such discussions with practical evidence from the literature.
>
> [R1] Feldman, Jon, et al. "Online stochastic packing applied to display ad allocation."
>
> [R2] Yuan, Shuai, Jun Wang, and Xiaoxue Zhao. "Real-time bidding for online advertising: measurement and analysis."
>
> [R3] Golrezaei, Negin, et al. "Learning product rankings robust to fake users."
>
> Regarding Weakness 2: We would like to respectfully respond to the reviewer’s comment “the technical contributions largely synthesizes known approaches, so the marginal technical contribution is not extremely high”. The most relevant previous theoretical work on this topic may be [R4], which studies the best-of-many-world setting for online allocation problems. The key difference is that the reward function and the constraint functions are known in [R4] before making the decision (i.e. [R4] studies the full information setting), while in this paper we study the bandit setting where the reward and constraint functions are unknown before a decision is made. This makes the analysis much more difficult than that of [R4], as we need to carefully handle unknown per-round constraint violation (before making decisions) to satisfy long-term constraints, while not allowing regret to decay too much. Our proposed algorithm requires much more complex design and analyses than that of [R4], which simply employs a mirror descent approach. We would also like to point out that although our proposed algorithm is inspired by well-established approaches, it is interpretable and implementable and thus can be easily adopted in practical setups. We also believe that our proposed algorithm can inspire future works to develop more complex algorithms and improve performance guarantees (to match with lower bounds for the full-information setting).
>
> Finally, as the reviewer describes, another key contribution of the paper is the modeling aspect, as to the best of our knowledge this is the first work that studies a universal algorithm that yields reasonable performance in various non-stationary autobidding setups under bandit feedback.
>
> [R4] Balseiro, Santiago R., Haihao Lu, and Vahab Mirrokni. "The best of many worlds: Dual mirror descent for online allocation problems."

---

> > ### Comment · Reviewer_hupT · 2023-08-11
> >
> > Dear authors,
> >
> > Thank you for the thorough response.  Your point about the difficulties in extending to a fully game-theoretic setup is well-taken.  I am happy to hear about your plan to include a discussion about these connections.
> >
> > Your response about the relationship to [R4] was very helpful.  I agree with your assessment of the additional challenges; your note about the bandit feedback setup is especially clear in this regard.  I agree that the practicality/implementability afforded by this setup is a strength.

---

### Official Review · Reviewer_g73L · 2023-07-10

**Soundness:** 3 good
**Presentation:** 3 good
**Contribution:** 3 good
**Rating:** 6
**Confidence:** 3

**Summary:**

The paper proposes a universally constrained online learning framework for ad procurement in non-stationary autobidding worlds. The paper makes contributions to the field by addressing the challenges of ad procurement in non-stationary autobidding worlds and developing a unified algorithm that can perform well in autobidding world while satisfying long-term constraints.


**Strengths:**

- The paper addresses the challenges of ad procurement in non-stationary autobidding worlds and develops a unified algorithm that can perform well in autobidding world while satisfying long-term constraints.
- The paper makes contributions to the field by developing an algorithm that yields good performance guarantees under different procedures.
- The paper is well-written and clearly presents the problem and the proposed solution.

**Weaknesses:**

- The paper does not have experiments to validate the theoretical findings.
- The paper could provide more insights into the practical implications of the proposed algorithm and how it can be applied in real-world settings.


**Questions:**

What are the challenges with applying the algorithm to obtain experimental results?

**Limitations:**

Yes.

---

> ### Author Rebuttal · Authors · 2023-08-10
>
> We thank the reviewer for the constructive and valuable feedback.
>
> Regarding Weakness 1 and Question 1 on experimental results: we agree that having experimental results would strengthen our paper’s key messages as well as contributions. We did not include experimental results in our paper due to space constraints, and we will add relevant discussions/results in our revision of this paper. We would like to point out that there are no major challenges in applying our algorithm to real or synthetic data to obtain experimental results, as the proposed algorithm is quite clean and easy to implement in practice. We would also like to mention that the main contributions of the paper lies in the modeling aspect (for autobidding under realistic non-stationary and limited information environments) as well as theoretical results.
>
> Regarding Weakness 2 on practical implications of our proposed algorithm: Online advertisers nowadays face a large array of advertising platforms such as search engines, social media platforms, web publisher display etc. Determining how to set advertising goals in each of these platforms (e.g. allocating total budget across different campaigns on a platform or setting target cost-per-click etc), especially in non-stationary environments due to changing user behavior, competition etc., becomes essential for online advertisers to optimize ad conversion outcomes. Our proposed algorithm presents a rigorous methodology that helps online advertisers set advertising goals in these ad platforms under non-stationary markets. As described above, we will also include experimental results in our revision to showcase the practicality of our proposed methodology for real-world settings.

---

### Decision · Program_Chairs · 2023-09-21

**Decision:**

Accept (poster)

**Comment:**

Overall, reviewers are all positive about the submission. In particular, the reviewers think the paper has an interesting modeling and conceptual to the line of research in auto-bidding.

On the other hand, reviewers do mention some weaknesses, for example (1) the paper does not have empirical studies, and it's not clear how well the theoretical findings fit practical applications (2) technical contributions might be marginal. These prevent the paper to get a higher rating. Reviewers also listed some typos and authors should try to fix them before publication.